# An evaluation of flow-routing algorithms for calculating contributing area on regular grids

Alexander B. Prescott[1], Jon D. Pelletier[1], Satya Chataut[2], and Sriram Ananthanarayan[2]

[1]Department of Geosciences, The University of Arizona, 1040 East Fourth Street, Tucson, Arizona 85721–0077, U.S.A.
[2]BHP Technical Centre of Excellence and Legacy Assets, 6840 North Oracle Road, Ste 100. Tucson, AZ 85704, U.S.A.

*Correspondence to*: Alexander Prescott (alexprescott@arizona.edu)

**Abstract.** Calculating contributing area (often used as a proxy for surface water discharge) within a Digital Elevation Model (DEM) or Landscape Evolution Model (LEM) is a fundamental operation in geomorphology. Here we document that a commonly used multiple-flow-direction algorithm for calculating contributing area, i.e., D∞ of Tarboton (1997), is sufficiently
biased along the cardinal and ordinal directions that it is unsuitable for some standard applications of flow-routing algorithms. We revisit the purported excess dispersion of the MFD algorithm of Freeman (1991) that motivated the development of D∞ and demonstrate that MFD is superior to D∞ when tested against analytic solutions for the contributing areas of idealized landforms and the predictions of the shallow-water-equation solver FLO-2D for more complex landforms in which the water-surface slope is closely approximated by the bed slope. We also introduce a new flow-routing algorithm entitled IDS (in
reference to the iterative depth-and-slope-dependent nature of the algorithm) that is more suitable than MFD for applications in which the bed and water-surface slopes differ substantially. IDS solves for water flow depths under steady hydrologic conditions by distributing the discharge delivered to each grid point from upslope to its downslope neighbors in rank order of elevation (highest to lowest) and in proportion to a power-law function of the square root of the water-surface slope and the five-thirds power of the water depth, mimicking the relationships among water discharge, depth, and surface slope in
Manning's equation. IDS is iterative in two ways: 1) water depths are added in small increments so that the water-surface slope can gradually differ from the bed slope, facilitating the spreading of water in areas of laterally unconfined flow, and 2) the partitioning of discharge from high to low elevations can be repeated, improving the accuracy of the solution as the water depths of downslope grid points become more well approximated with each successive iteration. We assess the performance of IDS by comparing its results to those of FLO-2D for a variety of real and idealized landforms and to an analytic solution of
the shallow-water equations. We also demonstrate how IDS can be modified to solve other fluid-dynamical nonlinear partial differential equations arising in Earth-surface processes, such as the Boussinesq equation for the height of the water table in an unconfined aquifer.

## 1 Introduction

Contributing area is a key variable in many empirical equations for fluvial erosion and sediment transport rates. As such, calculating contributing area on a regular grid is a task performed in many digital elevation model (DEM) analyses that involve fluvial processes (e.g., Clubb et al., 2017) and during every time step of nearly every Landscape Evolution Model (LEM) (Tucker and Hancock, 2010). Although contributing area is often used as a proxy for surface water discharge, the complexity and computational expense of hydraulic models precludes their use in some applications (e.g., landscape evolution models, where a full hydraulic model would have to be performed for every time step in order to evolve the topography) in favor of simpler and more efficient methods ("flow-routing algorithms") that distribute area as a function of topographic slope and require fewer inputs than hydraulic models. In addition to calculating contributing area, flow-routing algorithms are used as reduced-complexity models for the fluvial transport of quantities besides contributing area. Pelletier et al. (2008), for example, used a flow-routing algorithm to simulate the fluvial transport of radioactive tephra following a hypothetical volcanic eruption through the then-proposed nuclear-waste repository at Yucca Mountain and Pelletier and Orem (2014) routed a DEM-of-Difference to obtain a map of volumetric fluvial sediment fluxes following a wildfire.

Calculating contributing area on a regular grid (i.e., one with equal distance between adjacent grid points in both directions) involves assigning each grid point in a DEM an area equal to $(\Delta x)^2$, where $\Delta x$ is the distance between adjacent grid points. Working in rank order from the highest to the lowest elevation, the $(\Delta x)^2$ values at each grid point are added to the areas routed to each grid point from upslope and partitioned to nearest-neighbor grid points downslope. In the simplest flow-routing algorithm, i.e., D8 or steepest descent (O'Callaghan and Mark, 1983), all of the incoming area to each grid point is partitioned to the nearest-neighbor grid point (including diagonals) with the steepest slope. Because the flow pathways in D8 are multiples of 45°, D8 yields unrealistic predictions for surface-water-flow pathways in any portion of a landform in which the slope aspect is not a multiple of 45°. Freeman (1991) developed one of the first multiple-direction flow-routing algorithms, MFD (in reference to the multiple-flow-direction nature of the algorithm). MFD partitions flow to downslope grid points in proportion to a power-law function of the slope in the direction of each downslope nearest-neighbor grid point. Freeman (1991) advised using an exponent of 1.1 in the power-law function of slope based on a trial-and-error minimization of the directional bias of his algorithm (his Fig. 4). He found that, for the outer-facing cone test case that approximates the divergent morphology typical of many hillslopes, values of $p$ higher than 1.1 bias flow towards the cardinal and ordinal directions of the grid, while values of $p$ smaller than 1.1 bias flow away from the cardinal and ordinal directions of the grid.

Tarboton (1997) argued that the MFD algorithm results in excessive dispersion, i.e., lateral spreading with increasing distance downslope. To address this problem, Tarboton (1997) developed D∞, which limits dispersion in part by partitioning contributing area to at most two nearest-neighbor grid points. Tarboton (1997) documented that D∞ predicts flow patterns with less error and bias than competing algorithms for the outer-facing cone and planar test cases (his Table 2).

Other flow-routing algorithms for use on regular grid DEMs have been developed, including variations on MFD and D∞ (e.g., Quinn et al., 1991; Qin et al., 2007; Seibert and McGlynn, 2007), algorithms based on two-dimensional flow tubes (Costa-Cabral and Burges, 1994) and decomposed flux vectors (Desmet and Govers, 1996), and algorithms proposed for specific types of terrain or land uses (e.g., Hyväluoma et al., 2013 Xiong et al., 2014). Seibert and McGlynn (2007) introduced the triangular multiple flow direction algorithm that extends D∞ by permitting flow to more than two neighboring nodes when appropriate on divergent terrain. Qin et al. (2007) adapt the MFD algorithms of Freeman (1991) and Quinn et al. (1991) by allowing the exponent on local topographic slope to vary as a function of the maximum downslope steepness. The purpose of this modification was to improve the performance of the MFD algorithm in steep areas as a larger value of the exponent results in greater concentration of the flow in the direction of steepest descent and reduced dispersion (Holmgren, 1994; Qin et al., 2007). Alternatively, established flow-routing methods have been combined and modified for specific use cases. For instance, Xiong et al. (2014) route flow using the MFD algorithm for convex portions of the landscape and D8 on convergent portions of the landscape, while Hyväluoma et al. (2013) developed an anisotropic routing algorithm that allowed for explicit representation of directionally variable flow path likelihoods resulting from tillage. Such approaches trade method generalizability for improved accuracy in their study areas.

Previous work has compared the abilities of these and other algorithms to realistically distribute flow across the landscape in a variety of terrains that include both synthetic and real-world topography. Single flow direction algorithms (e.g., D8 or the stochastic variants of Fairfield and Leymarie (1991)) have been widely found to be insufficient at reproducing realistic flow paths, especially over divergent terrain such as hillslopes or distributary surfaces (Erskine et al., 2006; Gallant and Hutchinson, 2011; Qin et al., 2013; Rieger, 1998; Wilson et al., 2007; Zhou and Liu, 2002). Multiple-flow-direction algorithms have been shown to produce similar results over planar to convergent terrain while their largest relative differences occur in areas with lower contributing area totals such as ridgelines (Erskine et al., 2006; Wilson et al., 2007; Wilson et al., 2008; Zhou and Liu, 2002). Authors have variously argued for the primacy of D∞ (Gallant and Hutchinson, 2011; Tarboton, 1997), its multiple-direction variant (Seibert and McGlynn, 2007), or the variable exponent MFD variant (Qin et al., 2013).

The sensitivity of flow-routing methods to grid orientation has been previously examined in a limited number of publications, although the issue of DEM grid orientation dependence has also been addressed for other topographic metrics (e.g., Zhou and Liu, 2004). Fairfield and Leymarie (1991) documented the inability of D8 to capture flow directions correctly when surfaces were not oriented with the grid. Rieger (1998) compared the contributing area predictions of D8 and MFD on inner- and outer-facing cones and concluded that MFD showed better invariance to rotation. Hyväluoma (2017) explicitly considered the impact of grid rotation on MFD results for varying values of $p$, finding that rotational invariance was at a maximum for values near 1 and steadily declined (i.e., became more grid-orientation dependent) as $p$ increased. These results generally support the conclusion of Freeman (1991) to use a value of $p$ equal to 1.1 to minimize orientation artifacts.

One limitation of D∞, MFD, and the other aforementioned flow-routing methods is that they route areas using the bed slope. Contributing area is most often used as a proxy for surface water discharge, which is driven by water-surface slope. Unrealistic flow-routing patterns can result if the water-surface slope and the bed slope differ substantially (e.g., Fig. 1 of Bernard et al., 2022). Recent theoretical advancements have shown that traditional flow-routing algorithms are solutions to a simplified conservation equation for overland flowing water (Bonetti et al., 2018; Chen et al., 2014; Gallant and Hutchinson, 2011; Hutchinson et al., 2013). In particular, multiple-flow-direction algorithms are equivalent to the two-point flux finite volume approximation of Manning's equation (Coatléven, 2020; Coatléven and Chauveau, 2024). In light of these developments and in consideration of the attention that flow dispersion has received in the literature, a flow-routing algorithm that incorporates water discharge as a function of flow depth and water-surface slope could serve to more accurately compute specific contributing area on DEMs.

To address this limitation, we developed a water-depth-dependent flow-routing algorithm entitled IDS (referring to the iterative depth-and-slope-dependent nature of the algorithm) that provides additional accuracy for applications in which the bed and water-surface slopes differ substantially. IDS solves for the water surface under steady hydrologic conditions by distributing the discharge delivered to each grid point from upslope to its neighbors downslope in proportion to a power-law function of the product of the square root of the water-surface slope and the five-thirds power of the water depth, mimicking the relationships among water depth, surface slope, and discharge in Manning's equation. In Section 2, we provide background information on a case study that motivated this project. In Section 3, we describe the methods used to compare existing flow-routing methods on idealized and real-world topography, define the new IDS flow-routing algorithm, and describe how IDS can be modified to solve other flow-related nonlinear partial-differential equations arising in Earth-surface processes (in this case, the Boussinesq equation for the height of the water table in an unconfined aquifer). In Section 4, we describe the results of the comparisons between flow-routing algorithms. We assess the performance of IDS by comparing its results to those of FLO-2D (O'Brien, 2009; see also O'Brien et al., 1993) for a variety of real and idealized landscapes as well as to an analytic solution of the shallow-water equations applied to an idealized channel (Delestre et al., 2013; MacDonald et al., 1997). In Section 5, we discuss the implications of these results and the potential advantages and limitations of the IDS algorithm.

## 2 Motivating Example

The work documented here began with the goal of predicting the likelihood of rilling/gullying on a relatively long (350 m) and steep (up to 0.4 m/m) hillslope in Pinal County, Arizona (Fig. 1a). A necessary step in predicting the likelihood of rilling/gullying on hillslopes is to predict the peak specific discharge of surface water flow associated with rainfall events. To estimate the peak specific discharge, we installed monitoring equipment on the hillslope illustrated in Figure 1 to measure rainfall and water discharges (Pelletier et al., 2024). We then developed empirical equations relating the specific surface water

discharge to contributing area and peak event rainfall intensity. We attempted to use the D∞ algorithm to predict the likelihood of rilling/gullying on this hillslope and nearby hillslopes with other aspects/orientations but quickly ran into a problem: we found that the extent to which contributing area is localized into microtopographic depressions in the lower portions of the
hillslopes is highly sensitive to hillslope orientation. For hillslopes oriented along multiples of 45° (i.e., the cardinal and ordinal directions), D∞ predicts specific contributing areas in the lower portions of the hillslopes that are more than a factor of 2 larger than similar hillslopes in the study that are oriented in other directions, despite no substantial or apparent difference in the length or nature of the hillslopes. Figure 1 illustrates this phenomenon by comparing the specific contributing area, *a* (defined as the contributing area per unit distance perpendicular to the flow direction), predicted by D∞ and MFD for a hillslope oriented
along the vertical direction to that of the same hillslope with its point cloud rotated 30° prior to rasterization. For the hillslope without rotation, D∞ predicts a maximum specific contributing area of approximately 2300 m (Fig. 1b). For the same hillslope rotated 30°, D∞ predicts a maximum specific contributing area of less than 1000 m. MFD, in contrast, predicts similar specific contributing area values for the original and the rotated DEM (Fig. 1c).

A conclusion that could be drawn from Figure 1 is that D∞ returns correct results for one or the other landform orientation and that the dependence of the specific contributing area predicted by D∞ on landform orientation could be mitigated by orienting the hillslope properly prior to flow routing. This potential conclusion has two limitations. First, nearly all landscapes have a range of slope aspects/orientations, hence applying a rotation to achieve maximum accuracy would be impractical for all but the most planar or topographically simple study sites. Second, we have no way of knowing which orientation yields more
accurate results for any hillslope other than for idealized cases that have analytic solutions. Rather than interpreting the results of Fig. 1 as implying that D∞ is more correct for some orientations than others, Figure 1 implies that D∞ produces results that are indeterminate by more than a factor of 2 for this case. Given the sensitivity of rilling/gullying to whether a threshold shear stress or specific contributing area is exceeded, such indeterminacy renders D∞ unsuitable for this application. In this paper, we revisit the relative performance of D∞ and MFD in light of the sensitivity of D∞ to landscape orientation.

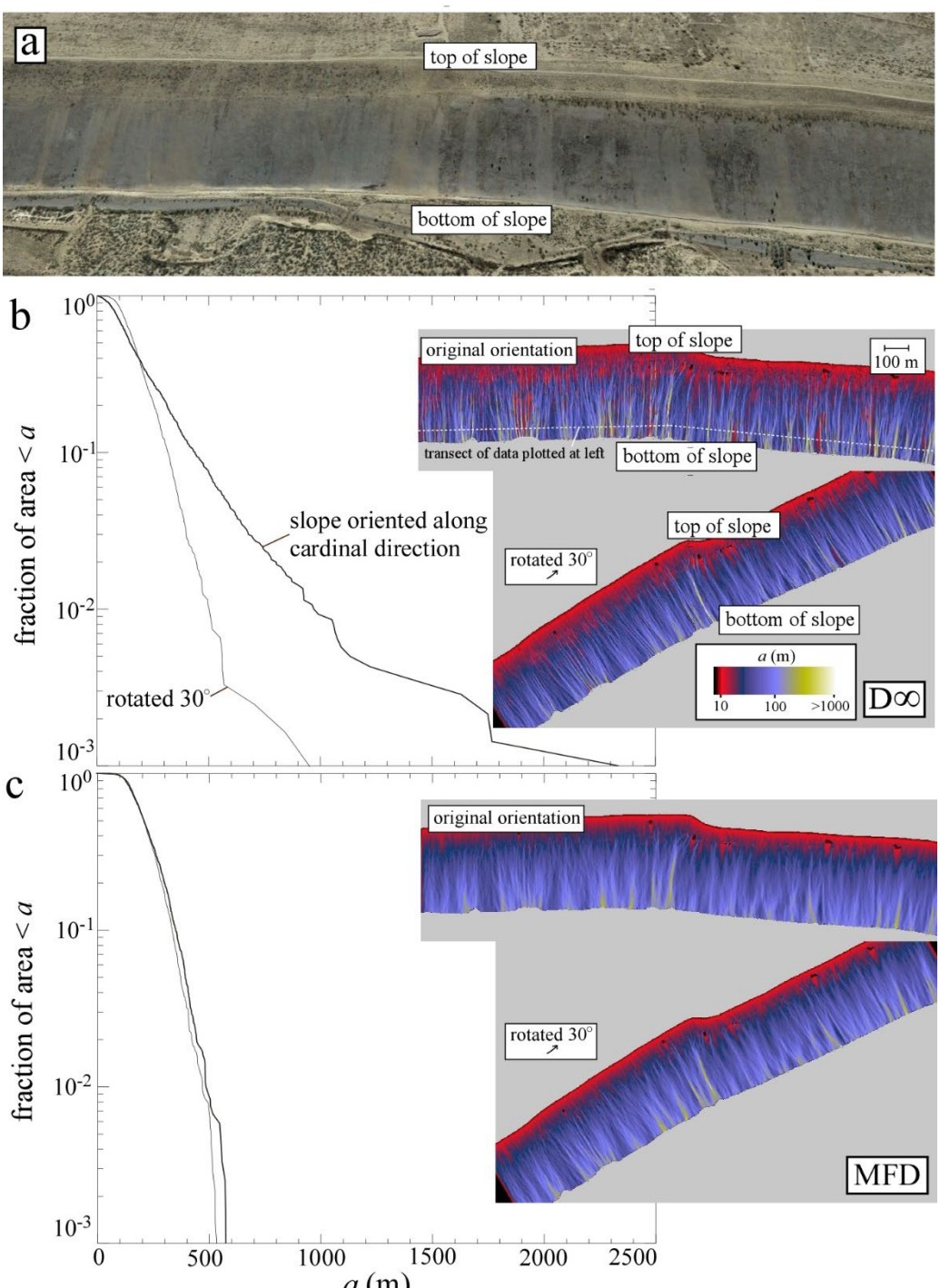

**Figure 1: Dependence of the specific contributing area, *a* (m), predicted by (b) D∞ and (c) MFD on the relative orientation of the hillslope to the cardinal and ordinal directions of the grid. (a) The monitored hillslope in Pinal County, AZ, USA, that motivated this work. (b) Rotating the hillslope by 30° results in more than a factor-of-two of difference in predicted *a* values using D∞. (c) The same rotation has almost no effect with MFD.**

## 3 Methods

### 3.1 Re-evaluation of D∞ and MFD for planar and conical slopes

This subsection details the comparisons made among D∞ and MFD and analytic solutions for the specific contributing area, $a$ (m), of idealized planar, outer-facing-cone, and inner-facing-cone test cases. We chose to compare D∞ and MFD in this study because of their widespread use in the community and because many of the other flow-routing algorithms commonly in use are derived from one or both of these algorithms.

The analytic solution for the specific contributing area of a plane is the straight-line distance parallel to the direction of flow from a given grid point to the upstream boundary (indicated by arrows in Fig. 2a). In this paper, we focus on the case of a plane oriented 30° relative to a cardinal direction to highlight algorithmic performance in cases in which the landform does not align with a cardinal or ordinal direction. The analytic solution for the outer-facing cone is:

$$a = \Delta x + \frac{r}{2} \tag{1}$$

where $r$ is the distance of the grid point from the center. There is some ambiguity about the correct value of $a$ as $r \to 0$ for the outer-facing cone. Mathematically, $a \to 0$ as $r \to 0$. In practice, however, every grid point is assigned an area equal to $(\Delta x)^2$ prior to flow routing when the specific contributing area is computed on a regular grid. Hence, the value of $a$ for any grid point that has no upslope neighbors is $\Delta x$. For this reason, we included a term in Eq. (1) that results in $a = \Delta x$ at $r = 0$.

The analytic solution for the inner-facing cone is:

$$a = \frac{\rho^2 - r^2}{2r} \tag{2}$$

where $0 < r \le \rho$ and $\rho$ is the radius of the cone.

The tests reported here were performed on grids with 101x101 points with $\Delta x = 1$ m. For the outer- and inner-facing cones, the cone center is located at grid point (51, 51) and comparisons between the analytic solution and the numerical results were made for areas with $r \le \rho = 50$ m. The central pixel was left out of the error assessment for the inner-facing-cone case due to the singularity at $r = 0$. The results presented in this paper using D∞ were obtained using version 5.3.7 of TauDEM (Tarboton, 2014).

### 3.2 Comparison of D∞ and MFD to FLO-2D for the landscape of Figure 1

Standard flow-routing algorithms do not involve discharge explicitly. Instead, each grid point is assigned a unit area equal to $(\Delta x)^2$ and that area is routed from upslope to downslope in a manner that depends on bed slope. These algorithms do not reference discharge because they implicitly assume that the bed slope and water-surface slope are equal.

As a prelude to relaxing the assumption that the water-surface slope is equal to the bed slope, we note that contributing area, $A$, may be defined as the ratio of the discharge, $Q$, (units of $L^3\,T^{-1}$) to a user-prescribed runoff rate, $R$, (units of $L\,T^{-1}$) under steady hydrologic conditions (i.e., steady, uniform flow and a time-invariant discharge in balance with a steady runoff rate):

$$A = Q/R \tag{3}$$

In this formulation, contributing area is explicitly a function of the discharge and the runoff rate. An advantage of this definition/formulation of contributing area is that it facilitates the computation of contributing area using the water-surface slope, which is the slope that drives surface water flow and thus imparts shear stress on the bed surface. Another advantage of this formulation is that a known discharge or flow depth entering the study area at an upslope boundary can be readily associated with a contributing area that can then be routed through the study area along with the area contributions from grid points within the DEM or LEM domain. One of the test cases considered in Sections 3.3 & 4.3 (i.e., the meandering channel) explicitly leverages this interrelationship between contributing area and discharge to accept incoming flow through an upstream boundary.

To evaluate the performance of the D∞ and MFD algorithms against the predictions of FLO-2D for the landform of Figure 1, we ran FLO-2D on the landform of Figure 1 with a constant, uniform runoff rate, $R$, for two hours. This simulation time was sufficient to produce a hydrologic steady state for $R$ ranging from 10 to 1000 mm hr$^{-1}$. The unit discharge was then converted to an equivalent specific contributing area using Eq. (3). We then compared the predictions of D∞ and MFD to those of FLO-2D using the cumulative frequency-size distributions of $a$ values along a contour located near the bottom of the hillslope.

**3.3 IDS**

IDS solves the steady state mass-conservation equation:

$$\nabla \cdot \mathbf{q} = R \tag{4}$$

where $\mathbf{q}$ is the unit discharge, given by the diffusive wave approximation of the shallow-water equations (Alonso et al., 2008):

$$\mathbf{q} = \frac{h^{5/3}}{n} |\nabla(b + h)|^{1/2} \hat{\mathbf{a}} \tag{5}$$

$h$ is water depth, $b$ is bed elevation, $n$ is Manning's n, and $\hat{\mathbf{a}}$ is a unit vector along the direction of the water-surface slope. IDS solves this system of equations for flow depth within a finite difference framework using a non-linear Jacobi iterative method (Ortega and Rheinboldt, 2000). A solution water surface is constructed incrementally from repeated grid traversals wherein grid points are solved sequentially according to a topological sort on water surface elevation (Heckmann et al., 2015; Klemetsdal et al., 2020), and discharge from a grid point is distributed among downstream neighbors using modified MFD partition weights (Table 1).

MFD assigns a unit area to each grid point equal to $(\Delta x)^2$ and, working in rank order from highest to lowest elevation, partitions the area entering each grid point from upslope to downslope grid points according to a power-law function of the bed slope:

$$f_{i,\text{MFD}} = \frac{S_i^p}{\sum_{i=1}^{8} S_i^p} \tag{6}$$

where $f_{i,\text{MFD}}$ is the fraction of the incoming contributing area that is partitioned to each of the 8 nearest-neighbors indexed with i, $S_i$ is the slope in each of the nearest-neighbor directions ($S_i = 0$ in eqn. (6) for any upslope grid points), and the default value of $p$ is 1.1.

IDS's approach to solving Eqs. (4) & (5) is a straightforward generalization of MFD that incorporates water-flow depth in addition to water-surface slope. IDS begins by using MFD and Eq. (3) to provide an initial guess for the discharge at every grid point. Manning's equation is then used to estimate the flow depth and water surface at every grid point using the discharge. The algorithm initializes the local discharge at each grid point to $R(\Delta x)^2$ and, working in rank order from highest to lowest elevation, adds the discharge routed to that grid point from upslope and partitions the discharge to downslope pixels according to:

$$f_{i,\text{IDS}} = \frac{\left(\frac{h_a^{5/3} S_i^{1/2}}{n_a}\right)^{2p}}{\sum_{n=1}^{8}\left(\frac{h_a^{5/3} S_i^{1/2}}{n_a}\right)^{2p}} \tag{7}$$

where $f_{i,\text{IDS}}$ is the fraction of the incoming discharge that is transferred to each of the 8 nearest-neighbor directions indexed by i, $S_i$ is the current best estimate of the water-surface slope in each of the nearest-neighbor directions, and $h_a$ is the weighted-average current best estimate of the water-flow depth and $n_a$ is the average Manning's $n$ value of the two grid points on the ends of the flow pathway between the central pixel and its nearest neighbor in each of the eight nearest-neighbor directions. The quantity in parentheses is the unit discharge in Manning's equation, assuming that the hydraulic radius can be approximated by the flow depth. The quantity in parentheses is raised to the power $2p$ to preserve the slope dependence that Freeman (1991) identified as resulting in optimal results for cones and planes.

IDS is iterative in two ways. First, when the grid is traversed from highest to lowest elevation and discharges are estimated at each grid point, the entire flow depth associated with the discharge is not added to each grid point all at once. Instead, IDS adds a fraction of the flow depth (equal to $1/N_a$) during each traversal of the grid, a process that is repeated $N_a$ times. The fractional flow depth is added as the difference between the new water surface and the current water surface, permitting incremental raising or lowering of the water surface as needed. This procedure facilitates lateral spreading of water flow in regions of unconfined flow. The second way that the IDS is iterative is that the entire procedure can be repeated $N_t$ times, using an improved estimate of the initial water-surface slope during each iteration. This repetition can yield improved accuracy for applications in which the water-surface and bed slopes differ substantially.

Table 1 summarizes the key steps of IDS. Note that if a better initial guess for the discharge is available (e.g., from a previous
time step within a LEM), that guess can be used instead of the results of MFD in step 1.

| |
|---|
| 1) Use the MFD algorithm to estimate discharge and water flow depth using Eq. (3).<br><br>2) Repeat $N_t$ times:<br><br> 2a) Repeat $N_a$ times:<br><br>  2a1) Assign a unit discharge to each pixel equal to $R(\Delta x)^2$. Add any discharges input through upslope boundaries.<br><br>  2a2) Compute water surface slope.<br><br>  2a3) Working from high to low elevations, partition the discharge from each grid point to its nearest neighbors using the fractions computed using Eq. (7).<br><br>  2a4) Compute the flow depth associated with the discharge at each grid point using Eq. 5.<br><br>  2a5) Add to the current estimate of the flow depth a fraction $1/N_a$ of the predicted flow depth.<br><br> End repeat loop beginning at 2a).<br><br>End repeat loop beginning at 2) |

**Table 1. Pseudocode for the IDS algorithm.**

Two subtleties associated with the IDS algorithm should be noted. First, the IDS algorithm performs a hydrologic correction
(using the priority-flood+ε algorithm of Barnes et al., 2014) implemented on the initial DEM and in between each traversal of
the grid to ensure that the water-surface slope does not fall below a user-prescribed minimum value. This is important because
water-surface slopes near zero can result in a prediction of unrealistically large flow depths when Manning's equation is used
to infer water-flow depth from discharge. Second, the simplest choice for calculating the $h_a$ term in Eq. (7) is to average the
water depths of the two grid points on either side of the flow pathway using equal weighting. However, we found that the
predictions of IDS match those of FLO-2D more accurately when the water depth of the grid point whose discharge is being
partitioned is weighted more than the water depth of the downslope grid point:

$$h_a = ch + (1 - c)h_i \qquad (8)$$

where $c \approx 0.8$ is used here because it yields results closest to those of FLO-2D (Section 4.3). Fiadeiro and Veronis (1977)
discuss how such weighted-mean schemes for finite difference approximations of steady state advection-diffusion-type
problems can improve the accuracy and/or convergence properties of the solutions.

IDS has six more parameters than either D∞ or MFD: the runoff rate $R$ (chosen to be consistent with the characteristic rainfall
event under consideration), Manning's $n$ (which can vary spatially to account for differences in surface roughness across the
DEM), the averaging parameter $c$, a minimum water-surface slope applied to prevent the water depth from becoming
unrealistically large when the discharge is converted to a flow depth, the number of additions of the fractional flow depth $N_a$,

and the number of complete water-surface constructions, $N_t$. Section 4 provides guidance on the choice of these parameters based on our experience with the test cases.

We tested IDS against the results of FLO-2D for the landform of Figure 1 and five idealized landscapes: the cones and plane of Figure 2, a low-order drainage basin, and a meandering fluvial channel. The low-order drainage basin is a useful test case because it demonstrates how IDS can resolve the variations in flow depth across a valley bottom (a capability that is essential for enabling LEMs to compute valley flow widths rather than requiring that a user prescribe them *a priori*). In this test case, the valley-bottom grid points cannot accommodate all of the discharge from the adjacent hillslopes, so the flow spreads laterally to occupy multiple grid points adjacent to the lowest grid point within each valley. The meandering channel example is a useful test case because it tests the ability of the IDS model to resolve the spatial variability of flow depths within a channel with bends and variable bed slopes and because it illustrates how flow through an upstream boundary can be accommodated.

We also tested IDS against an analytic solution for the shallow-water equations for steady, supercritical flow through an idealized 200-m-long rectangular channel with variable along-stream topography and width that includes a constriction and an expansion (Delestre et al., 2013). This case is pseudo-2D in that the analytic solution is not only depth-averaged but also width-averaged, i.e., topography and flow depth in the across-stream direction are constants. The channel width varies smoothly from about 10 m at the upstream and downstream boundaries to about 5 m at the constriction. Since the channel width is much wider than the grid spacing (we used 0.1 m for this value), flow through this channel is analogous to laterally confined sheet flow. We tested IDS on the 2D channel topography with a large number of iterations ($N_t = 2$, $N_a = 10,000$) to ensure convergence in the iterates, and averaged the solution flow depth across-stream to compare with the analytic solution.

### 3.4 Generalization of IDS to other partial differential equations in Earth-surface processes

In this subsection we describe how IDS can be modified to solve other flow-related steady state partial-differential equations using the 2D steady state Boussinesq equation as an example.

The 2D steady state Boussinesq equation quantifies the water table height in unconfined aquifers (Bear, 1972):

$$\nabla \cdot \big(h\nabla(b + h)\big) = -I/K \tag{9}$$

where $h(x,y)$ is the water-flow depth, $I$ is the recharge rate (units of L T$^{-1}$), and $K$ is the homogeneous and isotropic hydrologic conductivity (units of L T$^{-1}$). The boundary conditions used in the example application of this paper is $h = 0$ at any channel. Note that the Boussinesq equation is a conservation equation with a source term similar in form to Eq. (4). Specifically, Eqs. (9) and (4) & (5) are the same except that Eq. (9) has different power-law exponents among flux, water depth, and water-surface slope than Eqs. (4) & (5). The similarity in form of Eq. (9) and Eqs. (4) & (5) suggests that it should be possible to use the IDS algorithm, modified to partition the subsurface water flow entering each grid point to its nearest neighbors in proportion

to the product of $h$ and the water-surface slope $\nabla(b+h)$ raised to the power $p$, to solve the 2D steady state Boussinesq equation.

## 4 Results

### 4.1 Re-evaluation of D∞ and MFD for planar and conical slopes

Figure 2 illustrates the results D∞ and MFD for a plane oriented 30° counter-clockwise from south (Figs. 2a-2e), for the outer-facing cone (Figs. 2f-2j), and for the inner-facing cone (Figs. 2k-2o) test cases. Figure 2g is the key image in Figure 2 because it demonstrates that the relatively low dispersion of the D∞ along certain directions is a consequence of the tendency for flow to be biased along those directions ($a$ values are approximately 25% larger than the analytic solution along the cardinal and ordinal directions and lower everywhere else). MFD achieves a lower mean absolute error and bias than D∞ for all cases (Table 2). For the outer-facing cone, the error and bias obtained by MFD is nearly ten times lower than that of D∞. In Section 5 we discuss why the results presented here differ from those of Tarboton (1997).

| | MFD | | D∞ | | IDS | |
|---|---|---|---|---|---|---|
| | $\overline{\lvert a-a_a\rvert}$ (m) | $\overline{a-a_a}$ (m) | $\overline{\lvert a-a_a\rvert}$ (m) | $\overline{a-a_a}$ (m) | $\overline{\lvert a-a_a\rvert}$ (m) | $\overline{a-a_a}$ (m) |
| **Plane oriented 30°** | 3.55 | 1.28 | 7.51 | -7.50 | 3.65 | 2.80 |
| **Outer-facing cone** | 0.33 | 0.25 | 2.75 | -2.62 | 1.22 | 1.22 |
| **Inner-facing cone** | 2.24 | 2.17 | 6.40 | -2.79 | 2.32 | 2.00 |

Table 2. Performance of MFD, D∞, and IDS for the planar and conical test cases, as quantified using the mean absolute error $\overline{\lvert a-a_a\rvert}$ and mean bias $\overline{a-a_a}$, where $a_a$ is the analytic solution for specific contributing area.

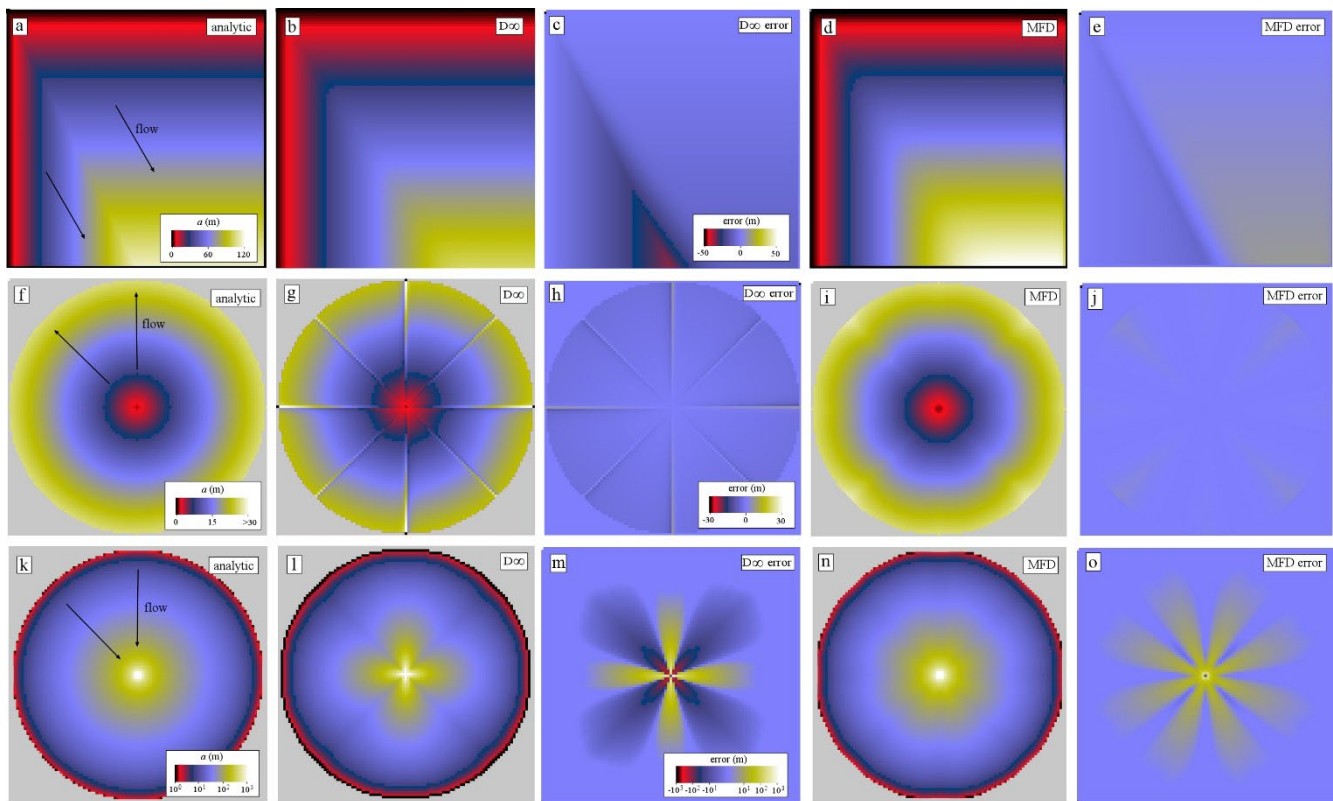

**Figure 2: Color maps of specific contributing area, *a* (m), and its error for D∞ and MFD for the (a)-(e) planar, (f)-(j) outer-facing cone, and (k)-(o) inner-facing cone test cases. The maps are arranged vertically for the three cases to facilitate comparison: (a), (f), and (k): analytic solutions; (b), (g), and (l): predictions using D∞; (c), (h), and (m): error using D∞; (d), (i), and (n): predictions using MFD; and (e), (j), and (o), error using MFD.**

Figure 3 complements Figure 2 by plotting the cumulative distribution of *a* values for each of the cases illustrated in Figure 2. Figure 3a demonstrates that, for the plane oriented 30° from the nearest cardinal direction, D∞ underpredicts the larger *a* values by approximately 20% while MFD overpredicts *a* values by approximately 10%. Figure 3b demonstrates that for the outer-facing cone, D∞ overpredicts *a* values in a small portion of the grid by approximately 25% and underpredicts *a* values nearly everywhere else, while MFD overpredicts *a* values by less than 5% everywhere. Note that, for the inner-facing cone, we plotted

the absolute error (Figs. 2m & 2o) using logarithmic scales due to the large positive skew of *a* values (i.e., a few grid points near the center have *a* values that are more than an order of magnitude larger than *a* values in most of the rest of the grid). Figure 3c documents that D∞ predicts absolute errors that are approximately 5-10 times larger than the absolute error associated with MFD for this case.

**4.2 Comparison of D∞ and MFD to FLO-2D for the landscape of Figure 1**

Figures 4 & 5 compare the specific contributing area predicted by D∞ and MFD (the results of D8 are also shown for completeness) to the predictions of FLO-2D for three values of the runoff rate: $R = 10$, 100, and 1000 mm hr$^{-1}$. Figures 4d-4f illustrate that contributing area is a function of runoff rate and/or water depth, and that any algorithm that seeks to calculate specific contributing area in a manner that honors the water-depth-dependence of flow routing (and its associated dispersion) should be a function of $R$. While the flow patterns are a function of $R$, a visual comparison of the differences among Figs. 4d-4f indicates that the specific contributing area is only modestly sensitive to $R$ for this test case, i.e., as $R$ increases over two orders of magnitude, dispersion increases (i.e., deeper flows are more likely to be laterally unconfined), but only modestly so on this steep hillslope. The modest dependence on $R$ is also apparent in Figure 5, where the dashed lines are the cumulative frequency-size distribution of $a$ values along the contour located as shown in Figure 1b. D8 produces results that are wholly unrealistic for this hillslope. The D∞ algorithm also produces results that are inconsistent with those of FLO-2D for this case (Fig. 5). The MFD algorithm produces results that are most consistent with FLO-2D (Fig. 5). We conclude that MFD is more accurate than D∞ for this test case.

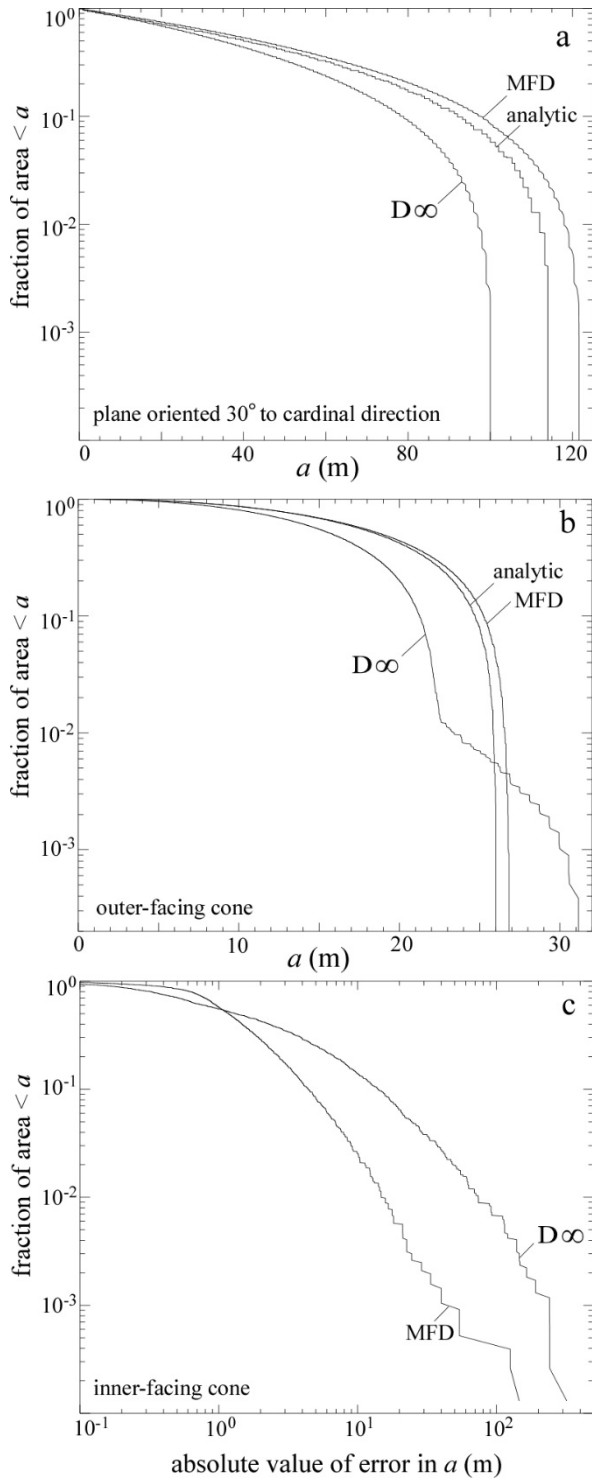

**Figure 3: Cumulative frequency-size distributions (i.e., the fraction of the area of the landform ≥ *a*) of (a) & (b) specific contributing area and (c) its absolute error for D∞ and MFD for the three test cases.**

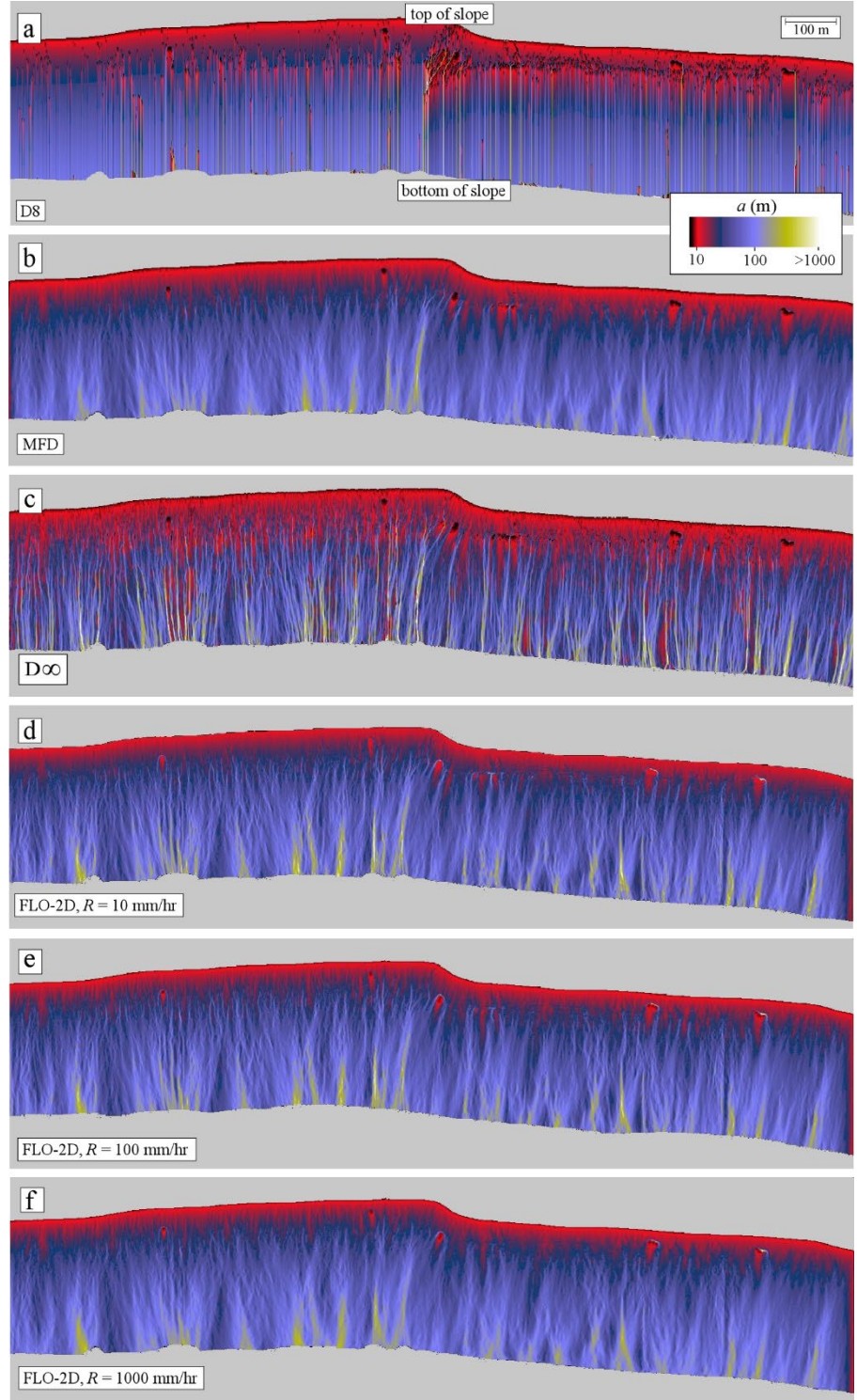

**Figure 4: Color maps of specific contributing area, *a* (m), for the hillslope in Figure 1 as predicted by (a) D8, (b) MFD, (c), D∞, and (d)-(f) FLO-2D for three values of the runoff rate, *R*.**

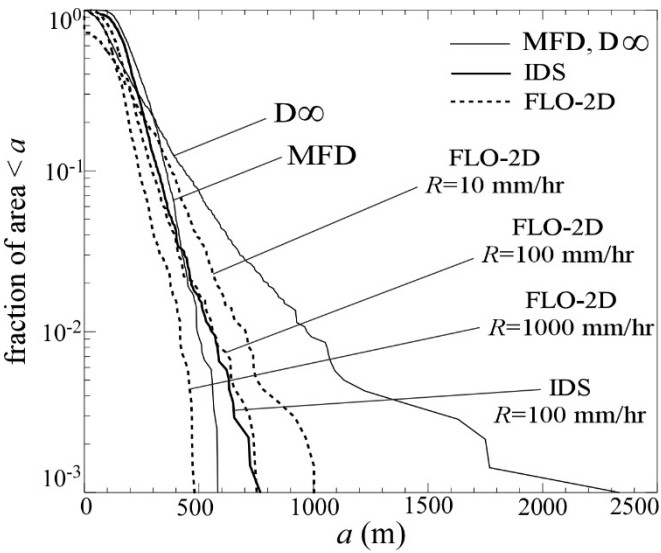

**Figure 5: Cumulative frequency-size distribution of specific contributing area, *a* (m), along the contour located in Figure 1b for the results in Figs. 4 and 7b.**

### 4.3 IDS

IDS predicts specific contributing areas for the planar and conical test cases with an accuracy comparable to that of MFD for the inner-facing cone case (Table 2 and Fig. 6). For the outer-facing cone and planar test cases, however, the results of IDS are inferior to those of MFD. Despite this weakness, IDS has an advantage in that it resolves flows within channels and valley bottoms, as the low-order valley and meandering channel examples presented in this section will demonstrate.

Figures 7a & 7b compare the predictions of IDS to those of FLO-2D for the hillslope of Figure 1. Figure 5 demonstrates that IDS predicts *a* values with a cumulative frequency-size distribution nearly indistinguishable from that of FLO-2D. Both IDS and FLO-2D result in essentially identical results when the DEM is rotated (results not shown because they are indistinguishable from those of Fig. 1c).

Figure 7c illustrates the results of IDS for the hillslope of Figure 1 using a DEM with $\Delta x = 0.5$ m. The results are similar to those of Figure 7b except that the higher resolution of the input data results in some finer detail in the flow pathways that are not present for the DEM with $\Delta x = 1$ m. This similarity between the results of Figure 7b & 7c provide confidence that we have implemented IDS (Prescott and Pelletier, 2024) in a manner that correctly converts between absolute and specific quantities as needed.

Figure 8 documents the dependence of the results of IDS on the parameter $c$ that controls the weighting of the local grid point relative to the downslope grid point when computing the water flow depth and Manning's $n$ value between grid points. Higher values of $c$ are associated with less topographic flow confinement. Figure 8d demonstrates that the value of $c$ that most closely matches the results of FLO-2D is 0.8. For all the other landscapes considered in this paper, the results are essentially independent of the value of $c$ within the range of reasonable values (i.e., 0.5 to 1).

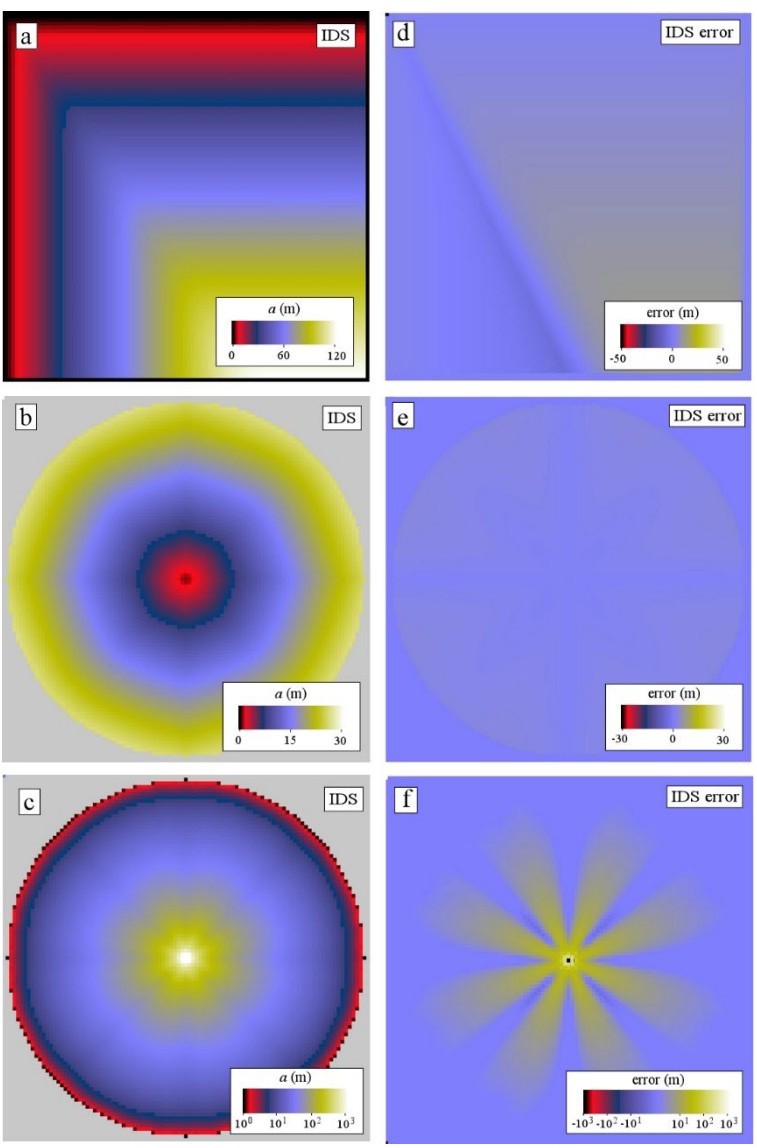

**Figure 6. Color maps of specific contributing area, $a$ (m), and its error for the IDS algorithm for the (a) & (d) planar, (b) & (e) outer-facing cone, and (c) & (f) inner-facing cone test cases.**

Figure 9 compares the results of IDS to those of FLO-2D for the case of a low-order drainage basin. IDS predicts flow depths that are nearly indistinguishable from those of FLO-2D for this test case. This low-order drainage basin is an illuminating example because it demonstrates how IDS can solve for the width of flow in addition to the depth where the wetted width of the valley is larger than the grid-point spacing. In this example, the accommodation space for surface water flow in the valley bottoms is sufficiently small that flow must spread out laterally into multiple grid points. If we were to use a flow-routing algorithm such as D∞ that is based on bed slope only, all of the flow would be localized into a single grid point (e.g., Bernard et al. 2022).

The inset diagram at the right of Figure 9b illustrates the dependence of the results on the number of flow depth additions implemented in IDS. For $N_a = 10$, the water-surface slope is slightly non-smooth across the profile, but this irregularity goes away for larger values of $N_a$. To determine the appropriate value of $N_a$, users should try multiple values to identify a value of $N_a$ above which the results change by less than the desired accuracy.

We evaluated the efficiency of IDS using this example for a range of grid sizes from 299x199 to 19,136x12,736. All of the simulations were run with identical parameters other than the number of grid points and the grid spacing. The computation time scales as $N\log_2 N$, where $N$ is the number of grid points (Fig. 10). This matches the theoretically optimal scaling of the two limiting functions used the algorithm, the priority-flood+ε depression-filling algorithm (Barnes et al., 2014) and the Quicksort algorithm used to rank the grid points from highest elevation to lowest (Sedgewick and Wayne, 2011). All other operations performed by the algorithm scale linearly.

Figure 11 compares the results of IDS to FLO-2D for the case of a meandering channel. The $R$ value was set to zero for this case and the water depth was assigned a value of 0.2 m in each of the grid points within the channel at the upslope boundary. Similar to Figure 9, IDS predicts flow depths that are visually indistinguishable from those of FLO-2D for this test case, except for a few grid points at or near the banks where there are some minor differences in predicted flow depths that could be related to how FLO-2D interpolates DEM points when developing a computational grid from an input DEM. This was the only example discussed in this paper in which it was necessary to use $N_t > 1$ to obtain results that were nearly indistinguishable from those of FLO-2D.

Figure 12 compares the flow depths predicted by IDS with the analytic solution of Delestre et al. (2013) for flow in a short channel of varying width. The discharge and depth at the upstream boundary is prescribed to be 20 m³ s⁻¹ (uniformly distributed among the upstream boundary grid points), and 0.503386 m. A uniform Manning's $n$ of 0.03 m$^{-1/3}$ s, grid spacing of 0.1 m, and an $R$ value of zero are also prescribed. Figure 13c shows that the water surface produced by IDS closely matches the analytical solution throughout the channel length with a mean residual of -0.057 m. The largest deviation occurs near the center of the channel profile where IDS underpredicts the analytic solution by 0.11 m, or 11.9% of the analytical flow depth. The IDS

solution is indistinguishable from the numerical solution to the full shallow-water equations applied to this case by Delestre et al. (2013) (their Fig. 9b).

The parameter values used in the applications of IDS illustrated in Figures 6-12 are summarized in Table 3. We chose $n =$ 0.035 for the cases that include fluvial valleys/channels, $n = 0.4$ m$^{-1/3}$ s for the hillslope case of Figs. 1, 4, 6, and 7, and $n =$ 0.03 m$^{-1/3}$ s for the case of Figure 12 to match the value used in the analytical solution. We adopted $n = 0.4$ m$^{-1/3}$ s for the hillslope case because Emmett (1971) recommended an approximate value of 0.5 m$^{-1/3}$ s for overland flow on hillslopes, but FLO-2D does not allow $n$ to be larger than 0.4 m$^{-1/3}$ s, so we adopted the 0.4 m$^{-1/3}$ s value as the closest value allowable in FLO-
2D to that of the Emmett (1971) recommendation. The results are not sensitive to the prescribed value of the minimum slope, provided that its value is smaller than the bed slope at all or nearly all locations. We choose 0.001 m m$^{-1}$ as the minimum slope for all the cases considered in this paper except for the low-order drainage basin case; that case required a lower value because it includes valley bottoms with bed slopes $< 0.001$ m m$^{-1}$. In all cases, Dirichlet boundary conditions (i.e., fixed elevation) were used for grid points along outflow boundaries or those with prescribed inflow conditions. While we believe that other boundary
conditions (e.g., prescribed flux and slope) could be implemented in IDS, ensuring that the problem specified by Eqs. 4 & 5 is well-posed with such boundary conditions requires additional research.

| Where presented | $R$ (mm hr$^{-1}$) | $n$ (m$^{-1/3}$ s) | $\Delta x$ (m) | $N_t$ | $N_a$ | $c$ | Minimum slope (m m$^{-1}$) |
|---|---|---|---|---|---|---|---|
| Figure 6 | 100 | 0.4 | 1 | 1 | 10 | 0.8 | 0.001 |
| Figure 7b | 100 | 0.4 | 1 | 1 | 10 | 0.8 | 0.001 |
| Figure 7c | 100 | 0.4 | 0.5 | 1 | 10 | 0.8 | 0.001 |
| Figure 8a | 100 | 0.4 | 1 | 1 | 10 | 0.5 | 0.001 |
| Figure 8b | 100 | 0.4 | 1 | 1 | 10 | 0.8 | 0.001 |
| Figure 8c | 100 | 0.4 | 1 | 1 | 10 | 1 | 0.001 |
| Figure 9b | 100 | 0.035 | 5 | 1 | 10 | 0.8 | 0.0001 |
| Figure 10 | 100 | 0.4 | 0.016-1 | 1 | 10 | 0.8 | 0.001 |
| Figure 11b | 0 | 0.035 | 10 | 3 | 100 | 0.8 | 0.001 |
| Figure 12b | 0 | 0.03 | 0.1 | 2 | 10,000 | 0.8 | 0.001 |

**Table 3. Parameters used in the example applications of the IDS algorithm.**

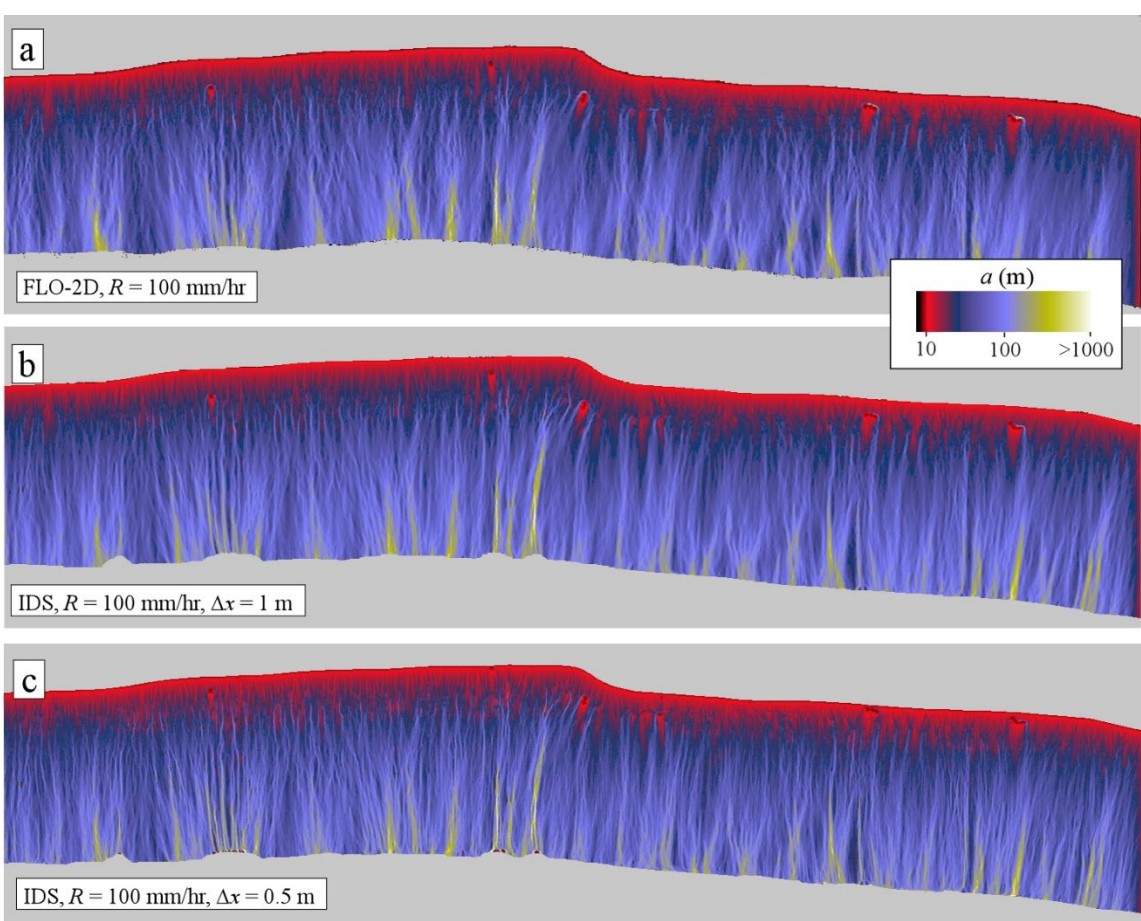

**Figure 7: Color maps of specific contributing area, *a* (m), for the hillslope pictured in Figure 1a as predicted by (a) FLO-2D and IDS using DEMs of different resolutions: (b) Δ*x* = 1 m and (c) Δ*x* = 0.5 m. All results correspond to *R* = 100 mm hr⁻¹.**

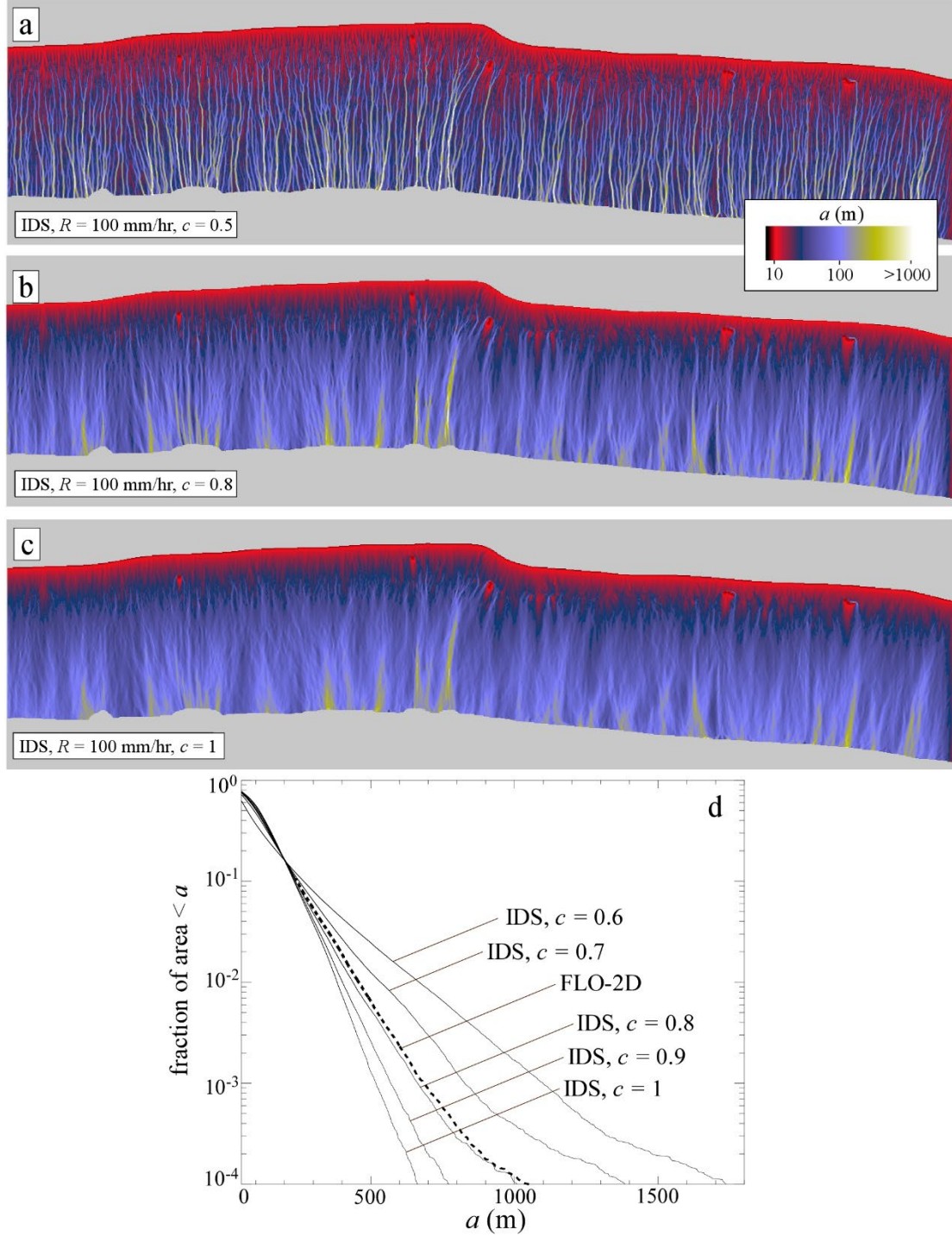

**Figure 8: (a)-(c) Color maps of specific contributing area, $a$ (m), for the hillslope pictured in Figure 1a as predicted by IDS for different values of the parameter $c$ quantifying the weighting of the local grid point relative to the downslope grid point. (a) $c = 0.5$, (b) $c = 0.8$, (c) $c = 1$. (d) Cumulative frequency-size distribution of $a$ (m) for a range of values of $c$ and for the results of FLO-2D.**

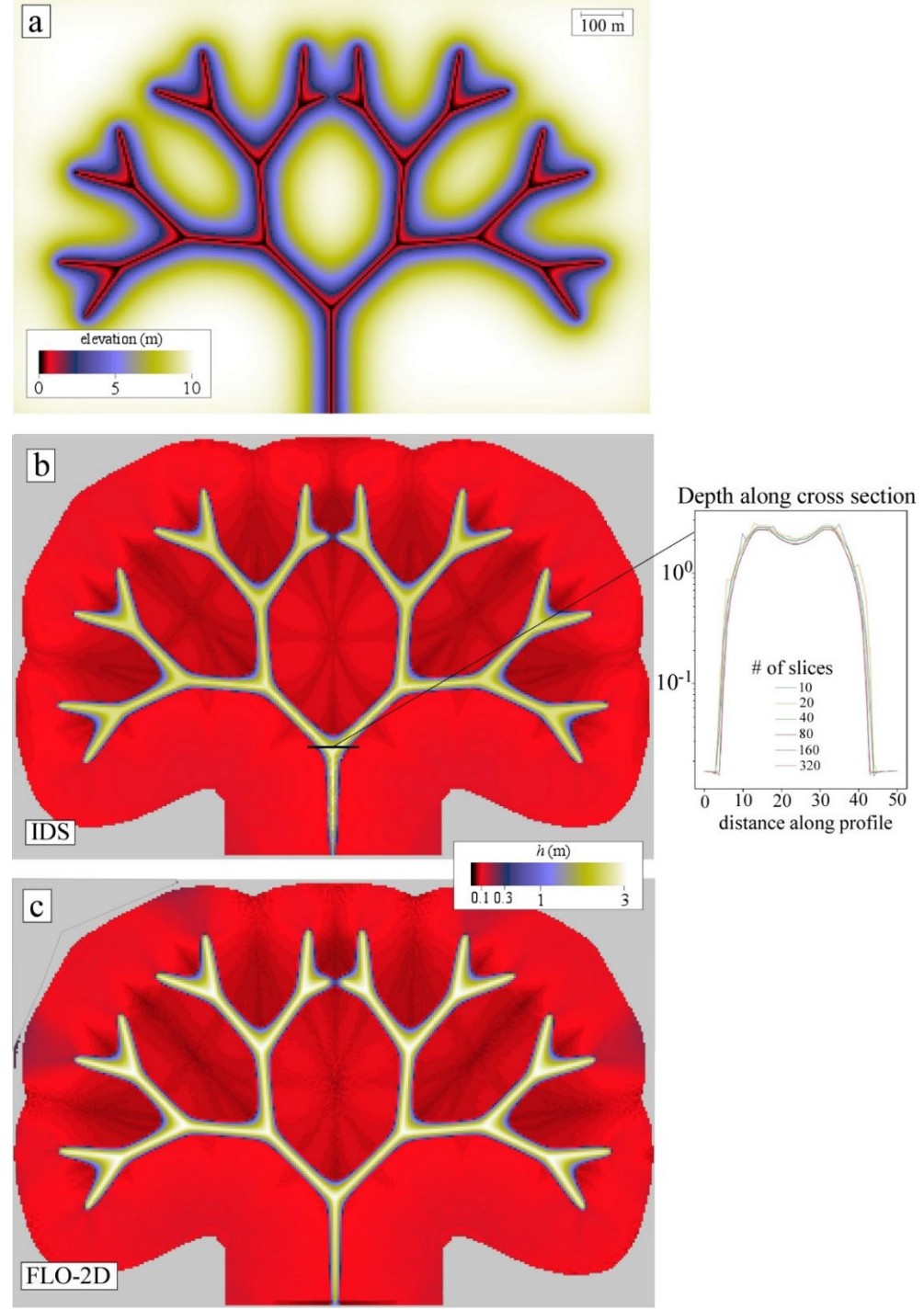

**Figure 9: Results of FLO-2D and IDS for a low-order drainage basin. (a) Color map of elevation. (b) & (c) Color maps of water-flow depth, $h$ (m), predicted by (b) IDS and (c) FLO-2D. The popout in (b) demonstrates convergence of the IDS solution water depth along a channel cross-section as the number of additions, $N_a$, is increased.**

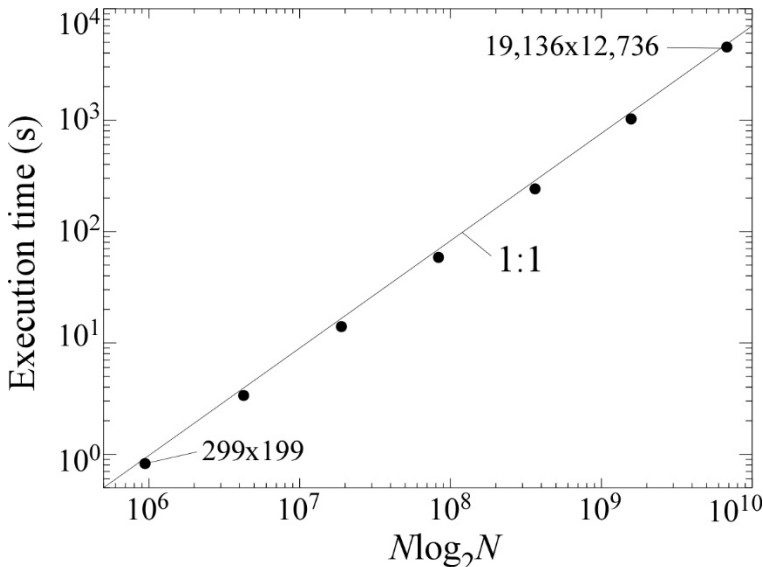

**Figure 10: Plot of execution time versus $N\log_2 N$, where $N$ is the number of grid points, for different resolutions of the low-order drainage basin of Figure 8.**

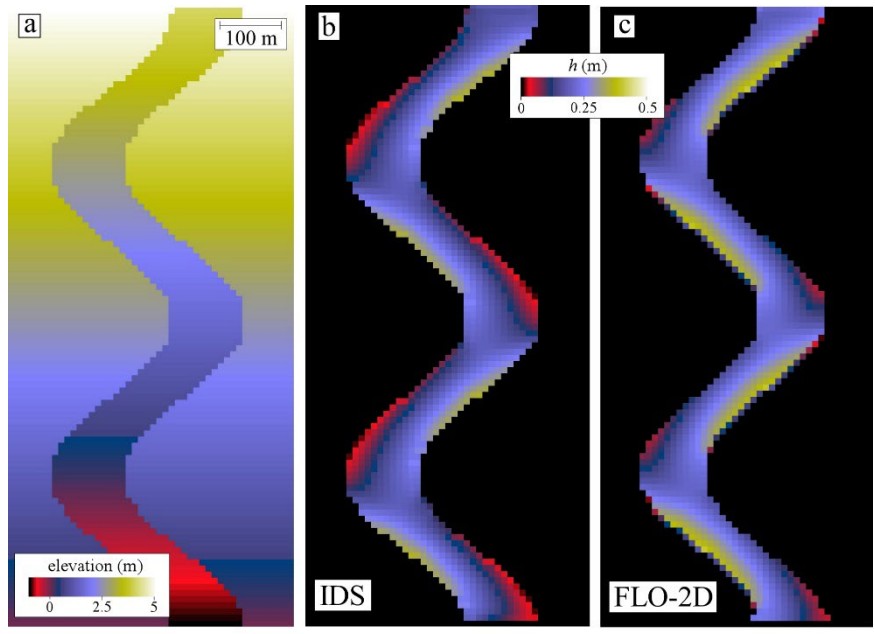

**Figure 11: Results of FLO-2D and IDS for the meandering channel example. (a) Color map of elevation of this test landform. (b) & (c) Color maps of surface water-flow depth, *h* (m), predicted by (b) IDS and (c) FLO-2D.**

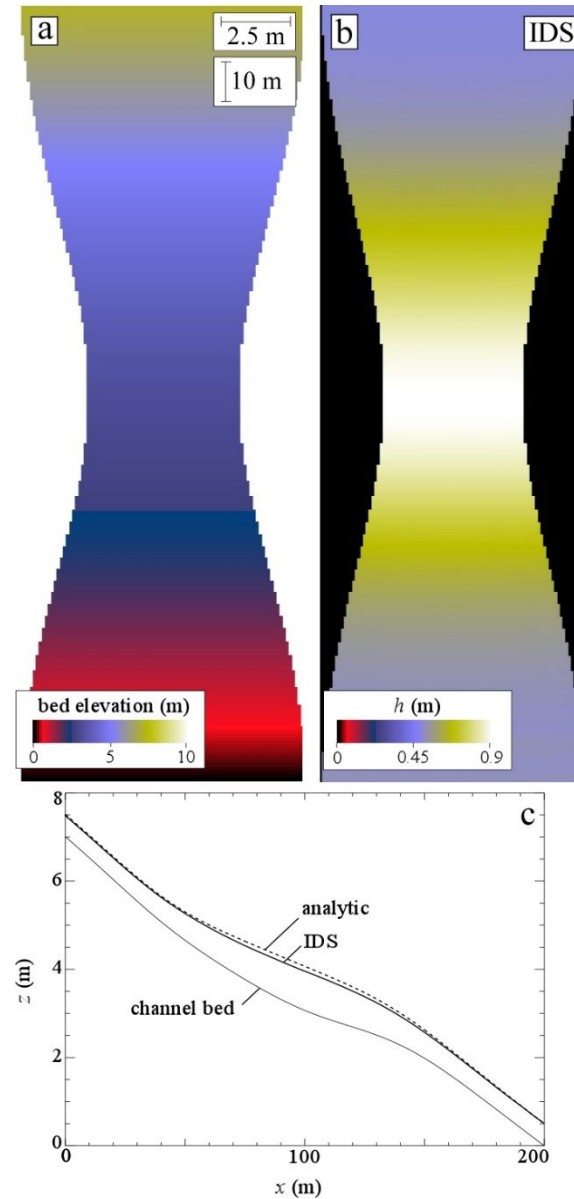

**Figure 12. Results of IDS applied to the pseudo-2D supercritical flow example of Delestre et al. (2013) and MacDonald et al. (1997). (a) Color map of elevation of the 2D channel bed. (b) Color map of the IDS solution for flow depth. (c) Comparison of the analytical solution of water-surface elevation to the IDS across-channel-averaged solution.**

## 4.4 Generalization of IDS to other partial differential equations in Earth-surface processes

Figures 13a & 13b illustrate the elevation of the water table in the vicinity of a straight channel. The analytic solution to Eq. (6) for this 1D case, assuming $b = 0$ and $I = 0$ for $x > L$, is obtained by integrating twice:

$$h = \sqrt{\frac{I}{K}(2Lx - x^2)} \tag{10}$$

That is, the flux of water in the saturated zone along the direction perpendicular to the channel increases linearly with distance from the left and right boundaries of the grid towards the channel. Figure 13b demonstrates that the numerical solution to Eq. (6) obtained using the IDS algorithm, $L = 500$ m, and $I/K = 0.001$ (unitless) is consistent with Eq. (10). Figures 13c & 13d illustrate the water-table elevation and the product of the water depth and water-surface slope (relevant because it is proportional to the water flux) for the more complex case of the water table in the vicinity of a meandering channel (where subsurface flow is localized towards the outer bends of the channel). Given that the seepage flux at a channel bank controls the bank stability to gravitational failure in alluvial channels (Cassagli et al., 1999; Simon and Collison, 2001) and that bank stability is an essential process is setting the hydraulic geometry of alluvial channels (Pelletier, 2021), this modification of IDS could prove useful in modeling the evolution of alluvial channels.

Table 4 summarizes the variable names, symbols, units, and default/typical values used in the paper.

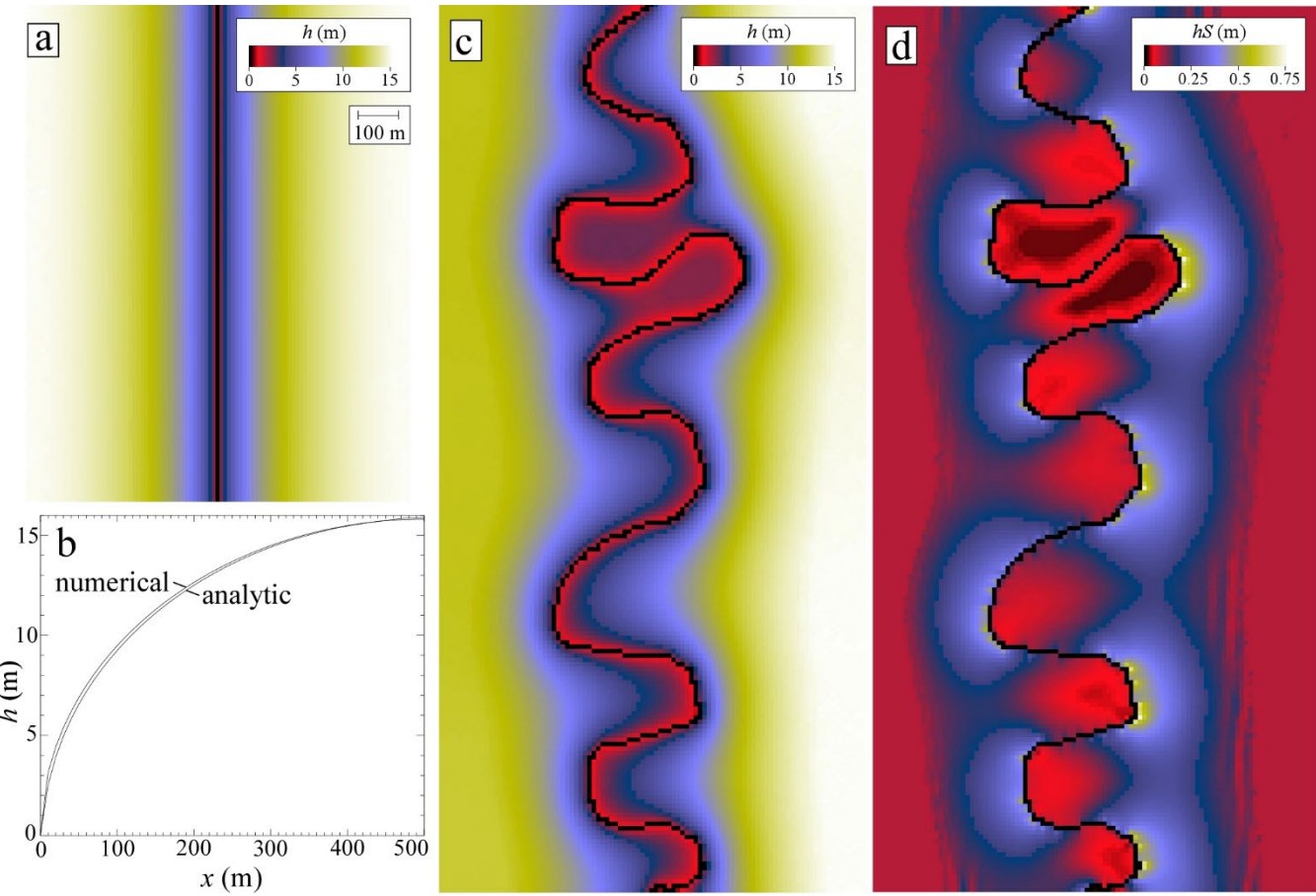

**Figure 13. Solutions to the Boussinesq equation for the water table elevation in the vicinity of a channel (where we assume *h* = *b* = 0). (a) Color map of water table depth in the vicinity of a straight channel. (b) Comparison of the solution mapped in (a) to the analytic solution. (c) & (d) Color maps of (c) water table depth and (d) the product of the water-table depth and water-surface slope in the vicinity of a meandering channel.**

## 5 Discussion

Although we devoted a substantial portion of this paper to documenting IDS, we wish to emphasize that MFD is adequate for many applications in which the water-surface slope is closely approximated by the bed slope. In such cases, MFD may be the preferred method given its simplicity and superior performance in matching analytic solutions for some idealized landforms. MFD is not suitable, however, for use in LEMs or DEM analyses that aim to resolve cross-sectional variations in surface water-flow depths in valley bottoms. For such cases, IDS or a similar depth-dependent flow-routing algorithm must be used despite the increased computational burden and complexity associated with such algorithms.

Qin et al. (2007) modified MFD to make the exponent *p* a function of the maximum downslope steepness. Their goal was to make flow more divergent in areas of gentle slopes and more convergent in areas of steep slopes. We believe that IDS is a

495 more direct and theoretically defensible approach to solving the problem of varying the degree of flow convergence/divergence based on local topography. Given that the exponent $p$ was introduced by Freeman (1991) not as a means to modify the degree of flow convergence/divergence but rather as a means of correcting for the tendency of flow to be biased towards or away from the cardinal and ordinal directions of the grid, varying $p$ to modulate convergence/divergence could come at the expense of introducing or amplifying a directional bias of the type documented in Figure 1. Indeed, Hyväluoma (2017) demonstrated

that the use of MFD with $p$ equal to 3 resulted in substantial grid orientation dependence, although this dependence could be counteracted with an intelligent weighting scheme. The dependence of results on grid orientation was at a minimum for $p$ equal to 1 and increased as the value of $p$ was increased. Considering that Qin et al. (2007) permit $p$ values of up to 10, we suspect that this method may also suffer from a grid orientation dependence.

Modifications of D∞ have also been proposed (e.g., Seibert and McGlynn, 2007; Shelef and Hilley, 2013). A reader might reasonably ask whether one or more of those modifications may have solved the bias documented in Figure 1, rendering a key motivation for this study moot. While we cannot be certain that the published literature contains no solution to the bias documented in Figure 1, the modifications to D∞ that we have examined have not solved the directional bias issue. For example, the modified version of D∞ proposed by Shelef and Hilley (2013) exhibits the same bias along the cardinal and

ordinal directions (see the panels related to directions 0° and 45° in Fig. 6 of Shelef and Hilley, 2013) that is apparent in Figure 2g. A similar bias is also apparent in Figure 4d of Seibert and McGlynn (2007). Past work has highlighted that multiple-flow-direction algorithms tend to differ the most along ridgelines in divergent topography (Erskine et al., 2006; Qin et al., 2013; Zhou and Liu, 2002). The results presented in Figure 1 demonstrate that a substantial dependence on grid orientation can result in large predicted differences in $a$ for convergent regions as well.

The data provided in this paper on the error and bias associated with D∞ and MFD differ from those of Tarboton (1997) (Table 2 in both papers) in part because Tarboton (1997) compared his results to analytic solutions for the contributing area downslope from a point source (i.e., examples illustrated in his Figs. 5-7). Shelef and Hilley (2013) highlighted the issue of whether flow-routing algorithms should be evaluated using a point source or equal-area contributions from each grid point. They concluded

that it is best to use a point source because "standardly used benchmarks that assume equal area contribution for each element in the landscape may offset errors in drainage area sourcing some points with errors from other points." We would share this concern for applications in which it is essential to accurately map which upslope grid points source which downslope grid points (e.g., predicting the area of contamination downslope from a localized source). For the standard application of flow-routing algorithms (i.e., calculating contributing area), however, we believe that it is necessary to test the algorithms in the

way that they are actually used (i.e., using equal area contributions from each grid point).

This study also differs from Tarboton (1997) in that we consider dispersion to be a necessary outcome of flow routing. The development of D∞ was motivated by the desire to minimize dispersion because it is "inconsistent with the physical definition

of upslope area, $A$" (Tarboton, 1997). Tarboton (1997) also concluded that "On a planar surface the dependence maps should
be straight lines perpendicular to the gradient" (i.e., dispersionless). In contrast, we propose that contributing area necessarily
involves dispersion because contributing area is a proxy for surface-water discharge and dispersion is present in all surface-
water hydraulic phenomena (e.g., Fischer, 1973). If the reader accepts that premise, it begs the question: what is the appropriate
amount of dispersion? We propose that, when flow-routing algorithms are used as reduced-complexity models for surface-
water hydraulics, the appropriate amount of dispersion is best identified by comparing the predictions of flow-routing
algorithms to those of hydraulic models, e.g., shallow-water-equation solvers such as FLO-2D and/or to analytic solutions to
the shallow-water equations.

IDS resides along a continuum of reduced-complexity algorithms for quantifying contributing area and/or surface water flow
that range from simple depth-independent algorithms (e.g., D∞ and MFD) to algorithms that approach the complexity of
solutions to the shallow-water equations (e.g., FLO-2D). Many models exist along this continuum, including LISFLOOD-FP
(Bates et al., 2010), FlowRCM (Liang et al., 1015), and FLOODOS (Davy et al., 2017). It is beyond the scope of this paper to
compare the results of IDS to these alternative approaches, but it is important to motivate the use of IDS by providing some
rationale for its use over alternative reduced-complexity algorithms for surface-water flow routing. One potential advantage
of IDS over LISFLOOD-FP is that IDS solves for a steady state hydrologic condition governed by a single characteristic or
peak runoff rate. LISFLOOD-FP, in contrast, requires an input time series of runoff. One potential advantage of the IDS
algorithm over FlowRCM and FLOODOS is that IDS is deterministic while FlowRCM and FLOODOS achieve lateral
spreading using random walkers. Deterministic approaches are advantageous because they return the same result each time
they are performed. Finally, the scaling behavior of IDS appears to be superior to that of FLOODOS (i.e., execution time
increases as $N\log_2 N$, i.e., more slowly than the $N^{1.2}$ scaling reported by Davy et al., 2017). Lastly, as demonstrated in the
application of IDS to the Delestre et al. (2013) example case (Fig. 12), IDS can closely approximate an analytic solution of the
shallow-water equations in addition to matching the results of FLO-2D across a range of scenarios.

We have treated FLO-2D throughout this paper as the gold standard for surface-water flow routing. A reader might reasonably
ask why we do not simply advocate for the use of FLO-2D as a flow-routing algorithm for use in LEMs and DEM analyses.
One reason is that its source code is unavailable. Another reason is that FLO-2D implements approximations that can affect
its accuracy (e.g., if the Newton-Raphson step fails to find a solution to the dynamic wave equation after 3 iterations, FLO-2D
reverts to the diffusive-wave approximation (p. 14 of O'Brien, 2009)). We did not compare the results of the various flow-
routing algorithms to FLO-2D quantitatively in this paper because, while FLO-2D is a widely respected and applied model, it
has not (to our knowledge) been tested against an analytic solution to the shallow-water equations. The extent to which it
represents the best or most exact solution possible for the other cases studied in this paper is also unknown. As such, we relied
on qualitative visual comparisons to avoid misinterpreting minor quantitative differences between the predictions of FLO-2D
and the flow-routing algorithms investigated in this paper as errors in the flow-routing algorithms.

The IDS algorithm routes flow under the assumption that discharge occurs across the wetted width of the grid spacing, $\Delta x$.
While consistent with our motivation to compute specific contributing area in situations where flow spreads laterally over multiple pixels, LEMs are also commonly applied with coarse grid resolutions such that channel widths are smaller than the grid spacing (Tucker and Hancock, 2010). A parameterization scheme could be implemented within IDS to allow for subgrid-scale channel widths while retaining the flow width on hillslopes or other areas of sheetflooding as $\Delta x$ (e.g., Pelletier, 2010).

It is worth briefly discussing the situations in which the IDS algorithm is computationally fast and those in which it requires many iterations to achieve a high degree of accuracy. Convergence occurs quickly in steeply sloping landforms where the water-surface slope is everywhere similar to the bed slope. In these applications, we have observed little to no sensitivity to the initial conditions. On the other hand, domains with relatively small bed slopes, large backwater lengths, and/or entirely subcritical flow conditions can require thousands of iterations to build an accurate water-surface solution. The performance of the algorithm also improves when the initial guess is closer to the final solution in such cases. This issue is common to numerical solvers of the diffusive-wave approximation, i.e., the doubly non-linear and degenerate nature of the partial-differential equation leads to difficulties as the water-surface slope goes to zero (Alonso et al., 2008). In addition, it is perhaps not surprising that, given the top-down iterative nature of the IDS algorithm (i.e., discharge is partitioned in rank order from the highest to lowest elevation), convergence requires more iterations in gently sloping flow domains (where the downstream water surface holds greater influence over the upstream surface) than it does in steeply sloping domains. It may be possible to invert the direction of the algorithm so that iterations proceed from prescribed downstream conditions and work through the topography in reverse rank order, thus allowing the downstream conditions to directly affect the upstream water surface. This alteration of IDS has not been tested and provides an avenue for future research.

## 6 Conclusions

The mapping of contributing area (often used as a proxy for surface water discharge) within a Digital Elevation Model (DEM) or Landscape Evolution Model (LEM) is a fundamental operation in many hydrologic and geomorphic models/analyses. Here we documented that a commonly used multiple-direction flow-routing algorithm, i.e., D∞ of Tarboton (1997), is inherently biased along the cardinal and ordinal directions. We revisited the purported excess dispersion of the MFD algorithm of Freeman (1991) that motivated the development of D∞ and demonstrated that MFD predicts contributing areas that are similar to those of analytic solutions for idealized cases and of the shallow-water-equation solver FLO-2D for more complex landforms. We also introduce a new flow-routing algorithm entitled IDS that provides additional accuracy for applications in which the bed and water-surface slopes differ substantially. We assessed the performance of IDS by comparing the results to those of FLO-2D for a variety of real and idealized landscapes and to an analytical solution of the shallow-water equations. We also demonstrated how the IDS algorithm can be modified to solve other flow-related nonlinear partial-differential

equations arising in Earth-surface processes, such as the Boussinesq equation for the height of the water table in an unconfined aquifer.

| Variable | Symbol | Units | Default value(s) |
|---|---|---|---|
| Contributing area | $A$ | m$^2$ | |
| Specific contributing area | $a$ | m | |
| Analytic solution for specific contributing area | $a_a$ | m | |
| Unit vector along slope aspect | $\hat{\mathbf{a}}$ | | |
| Bed elevation | $b$ | m | |
| Weight applied to local grid point when computing averages between neighboring grid points | $c$ | | 0.8 |
| Distance between adjacent grid points | $\Delta x$ | m | 0.1,0.5,1 |
| Partition coefficients for MFD algorithm | $f_{i,\text{MFD}}$ | | |
| Partition coefficients for IDS algorithm | $f_{i,\text{IDS}}$ | | |
| Water flow depth | $h$ | m | |
| Average water-flow depth between adjacent grid points | $h_a$ | m | |
| Water flow depth of nearest neighbor grid point in the direction labeled by index $i$ | $h_i$ | m | |
| Infiltration rate | $I$ | $L\,T^{-1}$ | |
| Index of eight nearest neighbor grid points | $i$ | | 1-8 |
| Hydrologic conductivity | $K$ | $L\,T^{-1}$ | |
| Flow distance from start of aquifer to channel | $L$ | m | 500 |
| Number of grid points | $N$ | | |
| Number of complete water surface constructions | $N_t$ | | 1-3 |
| Number of additions | $N_a$ | | 10,100,10$^5$ |
| Manning's $n$ | $n$ | m$^{-1/3}$ s | 0.4, 0.035, 0.03 |
| Average Manning's $n$ of neighboring points | $n_a$ | m$^{-1/3}$ s | 0.4, 0.035, 0.03 |
| Exponent on slope in MFD algorithm | $p$ | | 1.1 |
| Runoff rate | $R$ | mm hr$^{-1}$ | 10, 100, 1000 |
| Radius of conical hillslope | $\rho$ | m | 50 |
| Surface water discharge | $Q$ | m$^3$ s$^{-1}$ | |
| Unit discharge between adjacent grid points | $\mathbf{q}$ | m$^2$ s$^{-1}$ | |

**Table 4. List of variables and their associated symbols, units, and default and/or typical values. Note that $L\,T^{-1}$ only is listed for the units of $I$ and $K$ because these quantities only appear as a ratio, hence it is unnecessary to specify units of length and time.**

*Author contributions* JDP acquired funding for the project and conceptualized the study. ABP and JDP developed the model code, ran simulations, analyzed results, and wrote and revised the manuscript. SA and SC managed the study and provided helpful feedback.

*Code and data availability* The codes and data used for this study are available in Prescott and Pelletier (2024).

*Competing interests* The authors declare they have no competing interest.

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
