# Peer review of "An evaluation of flow-routing algorithms for calculating contributing area on regular grids"

_EGUsphere, 2024_

## Author Comment (AC1)

In this document, reviewer comments are left-aligned and colored in black, and our responses are indented and colored in blue. All line numbers in our responses refer to our proposed revised manuscript (without track-changes) unless otherwise specified. The proposed revised manuscript is included (with tracked changes) following the detailed response to the reviewers.

**Comments by Reviewer 1, anonymous** https://doi.org/10.5194/egusphere-2024-1138-RC1

> On 1 August 2024, we wrote an initial reply to Reviewer 1's comment (https://doi.org/10.5194/egusphere-2024-1138-CC1). Here, we emphasize and expand upon the points made in our initial response and highlight specific changes made to the text to address the reviewer's concerns.

The work deals with a comparison of flow-routing algorithms for calculating surface flow contributing area on regular grids.

As it is the manuscript does not represent an interesting contribution. It fails in the state of the art description of overland flow models avalaible and concentrates in a sort of comparison between the aprporximate algorithms IDS and D8 in order to show the superiority of IDS. At the same time the hydraulic shallow water based FLO-2D is used as a reference solution. Taking into account the many contribiutions that overland flow simulation has received in the last 20 years the scope of the present work is very narrow.

> We think that the work presented in this manuscript represents a complete study and is relevant to a broad cross section of earth surface science disciplines. To clarify the purpose of our study, we begin with a summary of our primary contributions and conclusions:
>
> 1. We documented a grid orientation dependence of the widely used D$\infty$ flow-routing algorithm that biases flow along the cardinal and ordinal directions. To our knowledge, our study is the first to explicitly demonstrate this bias and the indeterminate output of D$\infty$ that it causes on certain terrain.
> 2. We re-evaluated the D$\infty$ and MFD flow-routing algorithms by comparing their results for specific contributing area against analytic solutions on synthetic topography examples and against the hydraulic model FLO-2D on real-world topography. Our comparison differs from past work in our use of a hydraulic model as a reference solution and in that we use equal area contribiutions for every grid point in our comparison whereas other studies have used a point source. We concluded that MFD performed better than D$\infty$ in the test cases that we considered.
> 3. We presented a novel flow-routing algorithm that iteratively solves the diffusive wave approximation of the shallow water equations and partitions contributing area according to a power function of Manning's equation unit discharge. We compared the results of IDS against MFD, D$\infty$, and FLO-2D for the same example cases, as well as some additional comparisons with FLO-2D on more complicated synthetic terrain examples and a comparison with an analytical solution for shallow water flow through a channel with varying topography and channel width. We concluded that IDS performs similarly to MFD in these cases, compares well with FLO-2D, and closely approximates the shallow water flow analytic solution.
> 4. Finally, we demonstrated that the IDS algorithm could be easily modified for application to other geophysical fluids problems by modifying the exponents on unit discharge. We demonstrated this potential using the Boussinesq equation for unconfined groundwater flow in the vicinity of a channel.

We address the reviewer's comments one at a time by first quoting their passage and then responding to their concerns.

The reviewer writes "It fails in the state of the art description of overland flow models avalaible…" We responded in our initial reply that our focus in this study was not on the state of the art in hydraulic overland flow models but the specific domain of reduced-complexity flow routing models that are used to compute contributing area (or specific computing area) on equally spaced rectangular grids (i.e., regular grids).

To help clarify this main point, we added the following sentence to the first paragraph of the introduction on Lines 32-36:

> "Although contributing area is often used as a proxy for surface water discharge, the complexity and computational expense of hydraulic models precludes their use in some applications (e.g., landscape evolution models, where a full hydraulic model would have to be performed for every time step in order to evolve the topography) in favor of simpler and more efficient methods ("flow-routing algorithms") that distribute area as a function of topographic slope and require fewer inputs than hydraulic models."

In addition, in response to the comments of Reviewer 2, we added considerable background discussion regarding the type of reduced-complexity flow-routing algorithms that we considered here (see our response to their comment for additional details).

Reviewer 1 continues "… and concentrates in a sort of comparison between the aprporximate algorithms IDS and D8 in order to show the superiority of IDS."

We wish to emphasize that the bulk of our work involves the comparison of three multiple-flow-direction routing algorithms: MFD (Freeman, 1991), D∞ (Tarboton, 1997), and our new algorithm IDS. We include results of the single-flow-direction routing algorithm D8 in one place, Figure 4a, only as a means to demonstrate the inadequacy of D8 in our test case.

We chose to compare IDS against MFD and D∞ for multiple reasons: (i) IDS uses modified MFD weights to partition discharge amongst downstream neighbor grid points; (ii) MFD, D∞, and variants that have been developed upon their methods are in very wide use across scientific fields, landscape evolution model experiments, and GIS applications (e.g., ESRI ArcGIS Pro offers D8, D∞, and the MFD variant of Qin et al. (2007) as the three options for determining flow directions in their Spatial Analyst toolbox); and (iii) exactly because D∞ is so widely used, we believe it is important for the community to be made aware of the type of grid orientation bias that we document in this work. To our knowledge, this directional bias towards flow along the cardinal and ordinal directions has not been previously documented in the literature.

Reviewer 1 closes this paragraph with "At the same time the hydraulic shallow water based FLO-2D is used as a reference solution. Taking into account the many contribiutions that overland flow simulation has received in the last20 years the scope of the present work is very narrow."

This work is based on the following proposition: methods for computing contributing area should reflect the underlying physical characteristics of flowing water which they are meant to approximate. If one accepts this premise, then it is natural to compare the predictions of flow-routing algorithms for contributing area to the predictions made by a hydraulic model that is more

fidelitous to fluid dynamics. Considering that IDS solves the diffusive wave approximation to the shallow water equations, and considering that FLO-2D solves the full shallow water equations, FLO-2D presents an appropriate level of sophistication to compare against as a reference solution as it is closer to the full set of equations that govern the motion of viscous fluid (i.e., the Navier-Stokes equations). Therefore, it is reasonable to assume that the solutions of FLO-2D will be more accurate than IDS, MFD, or D∞ on topography without analytic solutions for specific contributing area. Given that our comparisons between flow-routing algorithms and FLO-2D are primarily visual and not quantitative, we do not believe that using a more sophisticated hydraulic model will substantively add to the results or conclusions of our study.

The scope of this work is narrow in the sense that we consider reduced-complexity flow-routing algorithms on regular grids and not the much wider and very diverse field of hydraulic modeling. However, as previously mentioned, reduced-complexity flow-routing algorithms are in very wide use across the fields of geomorphology, hydrology, geography, ecology, and others that are concerned with the movement of water, mass, and/or energy across the landscape. For this reason, we believe that the scientific, academic, and professional communities will find value in this contribution.

The formal aspects are also worth mentioning as the authors do not follow a clear structure in the presentation of the methods and results.

We appreciate this opportunity to improve the clarity and readability of the manuscript.

Motivated by this comment and the comments of Reviewer 2, we decided to modify the organization of the manuscript by splitting section 1 into two separate sections. Formerly, section 1 was titled "Introduction and Motivating Example", whereas our updated draft has "Introduction" in section 1 and "Motivating Example" in section 2. All subsequent section numbers have been increased by one to reflect this change.

Our Methods and Results sections are organized in a one-to-one manner wherein each subsection of the Methods is matched by an equivalent subsection in the Results section:
- In subsection 3.1, we introduce the methods that we used to compare MFD and D∞ against analytic solutions on synthetic test cases, and we present the results of this comparison in subsection 4.1.
- In subsection 3.2, we describe our methods for comparing MFD and D∞ against FLO-2D for the real-world topography of our motivating example, and we present these results in subsection 4.2.
- In subsection 3.3, we define the IDS algorithm and describe the test cases used to document its performance. We present these results in subsection 4.3.
- In subsection 3.4, we suggest that the IDS algorithm can be easily modified to solve equations involving the fluid flow in other geophysical domains, and we present an example of this extension in subsection 4.4.

To help clarify the organization of the Methods and Results, we simplified the subsection header titles and ensured that they match their counterpart across both sections. Subsections 3.1 and 4.1 are now titled "Re-evaluation of D∞ and MFD for planar and conical slopes", 3.2 and 4.2 are now titled "Comparison of D∞ and MFD to FLO-2D for the landscape of Figure 1", 3.3 and 4.3 remain titled "IDS", and 3.4 and 4.4 are now titled "Generalization of IDS to other partial differential equations in Earth-surface processes".

Even though the superiority of the IDS technique is clear from tthe tests presented the overall manuscipt is far from having the requiere quality to be recommended for publication.

> We thank the reviewer for recognizing the value of the IDS algorithm and hope that our edits to the manuscript and replies here have alleviated their concerns.

**Comments by Reviewer 2, anonymous** https://doi.org/10.5194/egusphere-2024-1138-RC2

Dear editor, dear authors,

The submitted article entitled "An evaluation of flow-routing algorithms for calculating contributing area on regular grids" looks at the problem of flow-routing algorithms.for calculating contributing area. The theme of this article is perfectly in line with the scope of the target journal, as shown by the article on the same theme (including the use of a modified MFD algorithm) recently published in ESurf: Coatléven, J. and Chauveau, B.: Large structure simulation for landscape evolution models, Earth Surf. Dynam., 12, 995–1026, https://doi.org/10.5194/esurf-12-995-2024, 2024.

The proposed article compares existing flow routing algorithms (MFD and D∞) with a new one (IDS). The new method takes into account hydraulic elements, which is not the case with conventional approaches. The algorithms are applied to several test cases. These demonstrate the superiority of the proposed method. The proposed method is of interest to the community.

However, the authors don not do enough to point out what already exists and the difficulties and scientific obstacles that exist. I suggest a state of the art/review of existing routing algorithms (and to complete the bibliography), giving the advantages and limitations of the algorithms mentioned, and justifying the choice of MFD and D∞ as reference algorithms among other existing algorithms. This would highlight the noveties and achievements obtained with the new proposed algorithm (both in introduction and conclusion).

Here is an example of bibliographical references that can be cited. This list is by no means complete, and authors are free to cite other references that may be more relevant.

1) Rieger, W. (1998). A phenomenon-based approach to upslope contributing area and depressions in DEMs. *Hydrological Processes*, *12*(6), 857-872. https://doi.org/10.1002/(SICI)1099-1085(199805)12:6<857::AID-HYP659>3.0.CO;2-B

2 ) Qiming Zhou & Xuejun Liu (2002) Error assessment of grid-based flow routing algorithms used in hydrological models,

International Journal of Geographical Information Science, 16:8, 819-842, DOI:10.1080/13658810210149425

3) Erskine, R. H., Green, T. R., Ramirez, J. A., & MacDonald, L. H. (2006). Comparison of grid-based algorithms for computing upslope contributing area. *Water Resources Research*, *42*(9).https://doi.org/10.1029/2005WR004648

4) Wilson, J. P., Lam, C. S., & Deng, Y. (2007). Comparison of the performance of flow-routing algorithms used in GIS-based hydrologic analysis. *Hydrological Processes: An International Journal*, *21*(8), 1026-1044. https://doi.org/10.1002/hyp.6277

5) Seibert, J., & McGlynn, B. L. (2007). A new triangular multiple flow direction algorithm for computing upslope areas from gridded digital elevation models. *Water resources research*, *43*(4). https://doi.org/10.1029/2006WR005128

6) Wilson, J. P., AGGETT, G., Yongxin, DENG., & LAM, C. S. (2008). Water in the landscape: a review of contemporary flow routing algorithms. *Advances in digital terrain analysis*, 213-236.

7) Xiong, L., Tang, G., Yan, S., Zhu, S., & Sun, Y. (2014). Landform-oriented flow-routing algorithm for the dual-structure loess terrain based on digital elevation models. *Hydrological Processes*, *28*(4), 1756-1766. https://doi.org/10.1002/hyp.9719

8) Coatléven, J. (2020). Some multiple flow direction algorithms for overland flow on general meshes.*ESAIM: Mathematical Modelling and Numerical Analysis*, *54*(6), 1917-1949. https://doi.org/10.1051/m2an/2020025

We thank the reviewer for their suggestions and their considered review of the manuscript. We added multiple paragraphs of discussion to the Introduction using the provided citations, as well as 14 additional references which we list below. In this new content, we more fully describe the history of flow-routing algorithm development, variants of the two main flow-routing algorithms as well as some alternatives, past studies that have compared the performance of existing algorithms, and the theoretical connection between flow-routing algorithms that use topographic slope and the systems of equations that describe overland flowing water.

The citations suggested by the reviewer have been added to the References section and in the text at the indicated line numbers:

1. Lines 81, 90: Rieger (1998)
2. Lines 81, 83, 511: Zhou and Liu (2002)
3. Lines 80, 83, 511: Erskine et al. (2006)
4. Lines 81, 83: Wilson et al. (2007)
5. Lines 63, 65, 84, 504: Seibert and McGlynn (2007)
6. Line 83: Wilson et al. (2008)
7. Lines 65, 72: Xiong et al. (2014)
8. Line 102: Coatléven (2020)

We also added the following citations to the References section and in the text at the indicated line numbers:

1. Line 100: Bonetti, S., Bragg, A. D., and Porporato, A.: On the theory of drainage area for regular and non-regular points, Proceedings of the Royal Society A, 474, 20170693, https://doi.org/10.1098/rspa.2017.0693, 2018.
2. Line 100: Chen, A., Darbon, J., and Morel, J.-M.: Landscape evolution models: A review of their fundamental equations, Geomorphology, 219, 68–86, https://doi.org/10.1016/j.geomorph.2014.04.037, 2014.
3. Line 102: Coatléven, J., and Chauveau, B.: Large structure simulation for landscape evolution models, Earth Surface Dynamics, 12(5), 995–1026, https://doi.org/10.5194/esurf-12-995-2024, 2024.
4. Lines 63-64: Costa-Cabral, M. C., and Burges, S. J.: Digital Elevation Model Networks (DEMON): A model of flow over hillslopes for computation of contributing and dispersal areas, Water Resources Research, 30(6), 1681–1692, https://doi.org/10.1029/93WR03512, 1994.
5. Line 64: Desmet, P. J. J., and Govers, G.: Comparison of routing algorithms for digital elevation models and their implications for predicting ephemeral gullies, International Journal of Geographical Information Systems, 10(3), 311–331, https://doi.org/10.1080/02693799608902081, 1996.
6. Lines 79, 89: Fairfield, J., and Leymarie, P.: Drainage networks from grid digital elevation models, Water Resources Research, 27(5), 709–717, https://doi.org/10.1029/90WR02658, 1991.
7. Lines 80, 84, 100: Gallant, J. C., and Hutchinson, M. F.: A differential equation for specific catchment area, Water Resources Research, 47(5), W05535, https://doi.org/10.1029/2009WR008540, 2011.

8. Line 70: Holmgren, P.: Multiple flow direction algorithms for runoff modelling in grid based elevation models: An empirical evaluation, Hydrological Processes, 8(4), 327–334, https://doi.org/10.1002/hyp.3360080405, 1994.
9. Line 101: Hutchinson, M. F., Stein, J. L., Gallant, J. C., and Dowling, T. I.: New methods for incorporating and analysing drainage structure in digital elevation models, in: Proceedings of Geomorphometry, International Society for Geomorphometry, Nanjing, China, 16–20 October 2013, 2013.
10. Lines 65, 73: Hyväluoma, J., Lilja, H., and Turtola, E.: An anisotropic flow-routing algorithm for digital elevation models, Computers and Geosciences, 60, 81–87, https://doi.org/10.1016/j.cageo.2013.07.012, 2013.
11. Line 46: O'Callaghan, J. F., and Mark, D. M.: The extraction of drainage networks from digital elevation data, Computer Vision, Graphics and Image Processing, 28(3), 323–344, https://doi.org/10.1016/S0734-189X(84)80011-0, 1984.
12. Lines 81, 85, 511: Qin, C.-Z., Bao, L.-L., Zhu, A.-X., Hu, X.-M., and Qin, B.: Artificial surfaces simulating complex terrain types for evaluating grid-based flow direction algorithms, International Journal of Geographical Information Science, 27(6), 1055–1072, https://doi.org/10.1080/13658816.2012.737920, 2013.
13. Lines 63, 67: Quinn, P., Beven, K., Chevallier, P., and Planchon, O.: The prediction of hillslope flow paths for distributed hydrological modelling using digital terrain models, Hydrological Processes, 5(1), 59–79, https://doi.org/10.1002/hyp.3360050106, 1991.
14. Lines 87-88: Zhou, Q., and Liu, X.: Analysis of errors of derived slope and aspect related to DEM data properties, Computers and Geosciences, 30, 369–378, https://doi.org/10.1016/j.cageo.2003.07.005, 2004.

Given the added content to the Introduction section, we decided to split what was previously section 1, "Introduction and Motivating Example", into two sections, "1 Introduction" and "2 Motivating Example". We hope that this change will make clearer the role of our study and the flow-routing context in which it exists. All subsequent section numbers have been increased by one to accommodate this organizational change.

We added the following content to the Introduction section (Lines 62-105):

[revised manuscript text omitted]

We moved what would have been the final paragraph of the Motivating Example section to the end of the the Introduction and modified it to make the layout of our manuscript clearer to the reader. It now reads (Lines 105-118):

> "To address this limitation, we developed a water-depth-dependent flow-routing algorithm entitled IDS (referring to the iterative depth-and-slope-dependent nature of the algorithm) that provides additional accuracy for applications in which the bed and water-surface slopes differ substantially. IDS solves for the water surface under steady hydrologic conditions by distributing the discharge delivered to each grid point from upslope to its neighbors downslope in proportion to a power-law function of the product of the square root of the water-surface slope and the five-thirds power of the water depth, mimicking the relationships among water depth, surface slope, and discharge in Manning's equation. In Section 2, we provide background information on a case study that motivated this project. In Section 3, we describe the methods used to compare existing flow-routing methods on idealized and real-world topography, define the new IDS flow-routing algorithm, and describe how IDS can be modified to solve other flow-related nonlinear partial-differential equations arising in Earth-surface processes (in this case, the Boussinesq equation for the height of the water table in an unconfined aquifer). In Section 4, we describe the results of the comparisons between flow-routing algorithms. We assess the performance of IDS by comparing its results to those of FLO-2D (O'Brien, 2009; see also O'Brien et al., 1993) for a variety of real and idealized landscapes as well as to an analytic solution of the shallow-water equations applied to an idealized channel (Delestre et al., 2013; MacDonald et al., 1997). In Section 5, we discuss the implications of these results and the potential advantages and limitations of the IDS algorithm."

In addition, we removed one sentence and one phrase in 2 Motivating Examples section that incorrectly attributed a statement to the work of Hyväluoma (2017). We moved a reference to Hyväluoma (2017) to the discussion section that more accrately described their work, which we quote a bit further down. The statements that were removed from section 2 were:

> On Lines 73-75 of the original submission, "These results corroborate those of Hyväluoma (2017), who first documented the enhanced sensitivity of low-dispersion flow-routing methods such as D∞ to landform orientation."

> On Lines 87-88 of the original submission, we removed the final phrase of the closing sentence, "…first documented by Hyväluoma (2017) and corroborated by Figure 1."

To justify our choice of MFD and D∞ as reference algorithms among other existing algorithms, we added the following sentence to the first paragraph of section 3.1 (Lines 159-161):

> "We chose to compare D∞ and MFD in this study because of their widespread use in the community and because many of the other flow-routing algorithms commonly in use are derived from one or both of these algorithms."

We also expanded our Discussion using these new references. On Lines 499-503, we added:

> "Indeed, Hyväluoma (2017) demonstrated that the use of MFD with $p$ equal to 3 resulted in substantial grid orientation dependence, although this dependence could be counteracted with an intelligent weighting scheme. The dependence of results on grid orientation was at a minimum for $p$ equal to 1 and increased as the value of $p$ was increased. Considering that

Qin et al. (2007) permit $p$ values of up to 10, we suspect that this method may also suffer from a grid orientation dependence."

On Lines 511-514, we added:

"A similar bias is also apparent in Figure 4d of Seibert and McGlynn (2007). Past work has highlighted that multiple-flow-direction algorithms tend to differ the most along ridgelines in divergent topography (Erskine et al., 2006; Qin et al., 2013; Zhou and Liu, 2002). The results presented in Figure 1 demonstrate that a substantial dependence on grid orientation can result in large predicted differences in $a$ for convergent regions as well."

The paragraphs dealing with the analytical solution in section 2.1 is not clear, it should be improved.

We made several minor wording changes to Methods subsection 1 (now section 3.1) to hopefully improve the clarity of the writing. These changes are:

- We altered the introductory sentence to more clearly describe the test cases. It now ends:
  "…for the specific contributing area, $a$ (m), of idealized planar, outer-facing-cone, and inner-facing-cone test cases."
- We altered the introductory sentence of the second paragraph. It now reads:
  "The analytic solution for the specific contributing area of a plane is the straight-line distance parallel to the direction of flow from a given grid point to the upstream boundary (indicated by arrows in Fig. 2a)."
- We added a comma and replaced "in" with "to" in the subsequent sentence for additional clarity.

We would be happy to make further edits as deemed necessary by the reviewer and the editor to further improve the clarity of the section.

Some details are necessary concerning the way the equations (4) and (5) are solved (numerical method, discretization/scheme, ...).

The numerical methods to solve equations (4) and (5) are described in the paragraphs that follow the introduction of these equations as well as in the pseudocode of Table 1. To make this clearer, we added a summary of the solution methods immediately after the definition of equation (5):

"IDS solves this system of equations for flow depth within a finite difference framework using a non-linear Jacobi iterative method (Ortega and Rheinboldt, 2000). A solution water surface is constructed incrementally from repeated grid traversals wherein grid points are solved sequentially according to a topological sort on water surface elevation (Heckmann et al., 2015; Klemetsdal et al., 2020), and discharge from a grid point is distributed among downstream neighbors using modified MFD partition weights (Table 1)."

We removed a text block from later in this subsection that repeats much of this information. This block occupied Lines 185-190 in the original submission.

In reviewing this section, we also noticed an inaccurate description of the codebase as it is stored in the Zenodo archive and as described in Table 1. We refer to the solution method as a "non-linear Gauss-Seidel iterative method" when in fact our submitted implementation uses a non-linear Jacobi iterative solution method. This line was a holdover from past versions of the code that did use a non-linear Gauss-Seidel method. IDS works with both solution methods and produces results that

are indistinguishable from one another. We apologize for this error and have corrected the text as shown in the quoted text block above.

There are a few typos to correct and things to clarify.

Line 90 : description of figure 1, I have the feeling that there is no description of subfigure (a), it has to be checked. All figures' legends need to be checked. On right part of b and c, axis might be added to help to understand the orientation of the graphics.

We modified inset parts of Figure 1 subpanels (b) and (c) by adding a label indicating which of the hillslopes showed the original orientation and by adding an arrow and label to indicate which of the hillslopes had been rotated by 30°.

We added the following description of subpanel (a) to the caption of Figure 1:

"…directions of the grid. (a) The monitored hillslope in Pinal County, AZ, USA, that motivated this work. (b) Rotating…"

In the caption of Figure 2, we replaced an instance of the article "a" with "the" and two commas with semi-colons for consistency.

In the caption of Figure 3, we added spaces between "(a)", "&", and "(b)".

In the caption of Figure 4, we added "frequency-size" to be consistent with Figure 3.

In the caption of Figure 6, we added spaces between references to subpanels as in Figure 3.

In the caption of Figure 7, we fixed a reference to subpanel "(c)", and we added the letter "c" to the Figure subpanel.

In the caption of Figure 8, we fixed references to subpanels "(b)" and "(c)". We also added "frequency-size" to be consistent with Figure 3 and removed "specific contributing area," from the final sentence to avoid repeating a definition already introduced in the first sentence.

In the caption of Figure 9, we added a description of the popout image in subpanel (b):

"…The popout in (b) demonstrates convergence of the IDS solution water depth along a channel cross-section as the number of additions, $N_a$, is increased."

In the caption of Figure 11, we added spaces between references to subpanels as in Figure 3.

In the caption of Figure 12, we added a reference to MacDonald et al. (1997) as further described in a subsequent reviewer comment and response.

Line 104, concerning FLO-2D, if possible it would be fine to cite an article in addition to the reference manual.

We added a reference to O'Brien et al. (1993) as an in-text citation and to the References section:

O'Brien, J.S., Julien, P. Y., and Fullerton, W. T.: Two-dimensional water flood and mudflow simulation, Journal of Hydraulic Engineering, 119(2), 244–261, https://doi.org/10.1061/(ASCE)0733-9429(1993)119:2(244), 1993.

Line 105 It is good to quote Delestre et al.'s article, which describes the SWASHES library of analytical solutions (a kind of review of analytical solutions for free-surface flows), but the authors should also quote MacDonald et al.'s article, which is the source of the solution.

9) I. MacDonald, M. J. Baines, N. K. Nichols, and P. G. Samuels. Analytic benchmark solutions for open-channel flows. Journal of Hydraulic Engineering, 123(11):1041–1045, November 1997

> We thank the reviewer for this suggestion and added the citation to the text as suggested as well as to the caption of Figure 12, and we added this entry to the References section. We did not add an in-text citation to MacDonald et al. (1997) at other places where Delestre et al. (2013) is cited as in those cases, we are specifically referring to the solution obtained from the implementation of Delestre et al. (2013).

Line 115 "algorithmic performance in in cases ..." I think that the sentence needs to be checked.

> We fixed this sentence by removing the redundant use of "in".

Line 147 "Sections 2.3$_3$.3 ..." spaces need to be added.

> We added spaces as suggested.

Lines 152-153 "verified to be sufficient time for  ..." I think that the sentence should be reformulated.

> We changed this sentence by splitting the parenthetical phrase into a separate sentence. The section now reads:

>> "…with a constant, uniform runoff rate, $R$, for two hours. This simulation time was sufficient to produce a hydrologic steady state for $R$ ranging from 10 to 1000 mm hr$^{-1}$."

Line 171 "... Eqs. (4)&(5) ..." spaces need to be added. Some other spaces need to be added lines 254, 255, 282, 290, 321, 328 and 421.

> We added spaces surrounding "&" symbols at the places suggested by the reviewer, as well as on lines 330, 337, 367, 374, and 471, and in the captions of Figures 3, 6, 9, 11, and 13.

Would it be able to apply the algorithm to unsteady flow?

> It may be possible to modify the algorithm to unsteady flow, although this is not something that we have considered deeply to this point. As it is currently formulated, the IDS algorithm instantly propagates information (in the form of water discharge) from the highest elevation grid point to the outlet grid point(s) in a single grid traversal. That is to say that there is no consideration of flood wave celerity in the algorithm, and it would take substantial modification to implement such functionality (if possible) and to ensure numerical stability (e.g., satisfying a Courant condition).

> Our focus here has been on steady-state hydrologic conditions because our primary motivation for calculating specific contributing area is for use in landscape evolution modeling. In this context, simulation timesteps are commonly on the order of 1kyr or more, far longer than the timescales of minutes, hours, days, and/or weeks for flood events on earth. For these reasons, we view an extension of IDS to unsteady conditions as beyond the scope of this work.

Best regards.

> Thank you!

**Additional changes to the manuscript**

On line 30, we replaced "DEM" with "digital elevation model (DEM)" as this is the first time this acronym is used outside of the Abstract.

In Table 1, we replaced an erroneous reference to "2b" with the correct reference to "2a". This line of the pseudocode now reads "End repeat loop beginning at 2a)."

In Figure 10, we updated the chart to reflect the x-axis values as calculated using log base 2. The version submitted accidentally calculated these values using log base 10, in conflict with the x-axis label. This change merely translates the points and does not alter the results or conclusions.

[revised manuscript text omitted]